# Predicting resilience during the COVID-19 Pandemic in the United Kingdom: Cross-sectional and longitudinal results

Kate M. Bennett[1]*, Anna Panzeri[2], Elfriede Derrer-Merk[1], Sarah Butter[3], Todd K. Hartman[4], Liam Mason[5], Orla McBride[3], Jamie Murphy[3], Mark Shevlin[3], Jilly Gibson-Miller[6], Liat Levita[7], Anton P. Martinez[6], Ryan McKay[8], Alex Lloyd[5], Thomas V. A. Stocks[6], Gioa Bottesi[2], Giulo Vidotto[2], Richard P. Bentall[6], Marco Bertamini[2]

1 Department of Psychology, University of Liverpool, Liverpool, United Kingdom, 2 Department of General Psychology, University of Padova, Padova, Italy, 3 School of Psychology, Ulster University, Coleraine, United Kingdom, 4 Department of Social Statistics, University of Manchester, Manchester, United Kingdom, 5 Division of Psychology and Language Sciences, University College London, London, United Kingdom, 6 Department of Psychology, University of Sheffield, Sheffield, United Kingdom, 7 School of Psychology, University of Sussex, Brighton, United Kingdom, 8 Department of Psychology, Royal Holloway, University of London, London, United Kingdom

* kmb@liverpool.ac.uk

**Data Availability Statement:** The data can be found at: https://osf.io/v2zur/

## Abstract

Although the COVID-19 pandemic has impacted the psychological wellbeing of some people, there is evidence that many have been much less affected. The Ecological Model of Resilience (EMR) may explain why some individuals are not resilient whilst others are. In this study we test the EMR in a comparison of UK survey data collected from the COVID-19 Psychological Research Consortium (C19PRC) longitudinal study of a representative sample of the United Kingdom (UK) adult population and data from an Italian arm of the study. We first compare data from the third wave of the UK arm of the study, collected in July/August 2020, with data from an equivalent sample and stage of the pandemic in Italy in July 2020. Next, using UK longitudinal data collected from C19PRC Waves 1, 3 and 5, collected between March 2020 and April 2021 we identify the proportion of people who were resilient. Finally, we examine which factors, drawn from the EMR, predict resilient and non-resilient outcomes. We find that the 72% of the UK sample was resilient, in line with the Italian study. In the cross-sectional logistic regression model, *age* and *self-esteem* were significantly associated with resilience whilst *death anxiety thoughts*, *neuroticism*, *loneliness*, and *Post Traumatic Stress Disorder (PTSD) symptoms* related to COVID-19 were significantly associated with Non-Resilient outcomes. In the longitudinal UK analysis, at Wave 5, 80% of the sample was Resilient. *Service use*, *belonging to wider neighbourhood*, *self-rated health*, *self-esteem*, *openness*, and *externally generated death anxiety* were associated with Resilient outcomes. In contrast, *PTSD symptoms* and *loneliness* were associated with Non-Resilient outcomes. The EMR effectively explained the results. There were some variables which are amenable to intervention which could increase resilience in the face of similar future challenges.

**Funding:** The funders had no role in study design, data collection and analysis, decision to publish, or preparation of the manuscript.

## 1. Introduction

In early 2020 the severe acute respiratory syndrome coronavirus 2 (SARS-CoV-2), known as COVID-19, spread across the world. By March and April 2020 many countries, including the United Kingdom (UK), were entering strict lockdowns, restricting movement, enforcing work from home, and prohibiting household mixing. The depth and breadth of the restrictions led both researchers and policy makers to suggest that there would be a tsunami of mental health problems. Studies at the beginning of the lockdown did indeed demonstrate an increase in psychological distress [1, 2]. However, as time progressed the impact of the pandemic became more nuanced, with people showing varying trajectories in symptoms and responses [3–5].

In March 2020, the COVID-19 Psychological Research Consortium (C19PRC) was launched in the UK with the aim to investigate the UK population's psychological, social and political responses to the pandemic. The C19PRC is an international project [6] with parallel surveys in several countries including Italy [7, 8], Spain [9] and Ireland [10]. Surveys were administered online with quota samples stratified to represent the countries for age, sex and household income. One important research question that the C19PRC has addressed is the extent of the mental health impact. Shevlin et al. [5], using the first three Waves of the survey (March–July 2020), identified five classes of anxiety-depression trajectory: resilient, chronic, adaptive, deteriorating, and vulnerable. The patterns of anxiety-depression that emerged in the UK during the first several months of the pandemic had stabilised by the fifth Wave (W5) one-year follow-up [5]. Remarkably, just under 70% of the sample demonstrated resilience in terms of low anxiety, depression and stress symptoms at this time point.

Resilience has been conceptualised in a number of ways, including as a trait [11], as stability in wellbeing despite challenge [12], or as bouncing back from adverse life events [13]. We utilise Windle's [14] large scale concept analysis, and employ her definition of resilience: "Resilience is the process of negotiating, managing and adapting to significant sources of stress or trauma. Assets and resources within the individual, their life and environment facilitate this capacity for adaptation and *bouncing back* in the face of adversity. Across the life course, the experience of resilience will vary" (p. 163). Resilience differs from notions of good mental health in its central requirement for challenge or trauma, in the case of this study, living through the COVID-19 pandemic [15]. Other contexts relevant to resilience include natural disasters [16], bereavement [17], dementia [18], and war [19]. Cosco et al. [20] reviews different measurement approaches, and in this study we operationalise resilience as the *absence of both depressive and anxiety symptoms* (below the criteria for clinical caseness) *in the face of significant challenge*, namely the COVID-19 pandemic.

A full understanding of the factors that led to this resilience will lead to recommendations for policy and practice to promote a more resilient response to future pandemics and similar events. One model which may help us to understand resilient responses is the Ecological Model of Resilience (EMR) [21]. This model posits that the presence or absence of individual, community and societal resources leads to a resilient outcome or alternatively to a negative effect on wellbeing (see Fig 1). Individual factors include demographic, psychological and health-related resources. Community variables include family, social support, and social participation. Societal variables include health and welfare support, social policies, and the economy. This model has been found to be a useful analytical tool for explaining resilience in the context of caring for people with dementia [22], widowhood [23], chronic health problems [24] and poverty amongst older adults living in Colombia [25].

In relation to the pandemic, the EMR has recently been applied amongst people living in four regions of Italy, two in the north of the country (Lombardia and Veneto) and two in the south (Campania and Lazio) [26]. We defined resilience as conjoint low levels of depressive

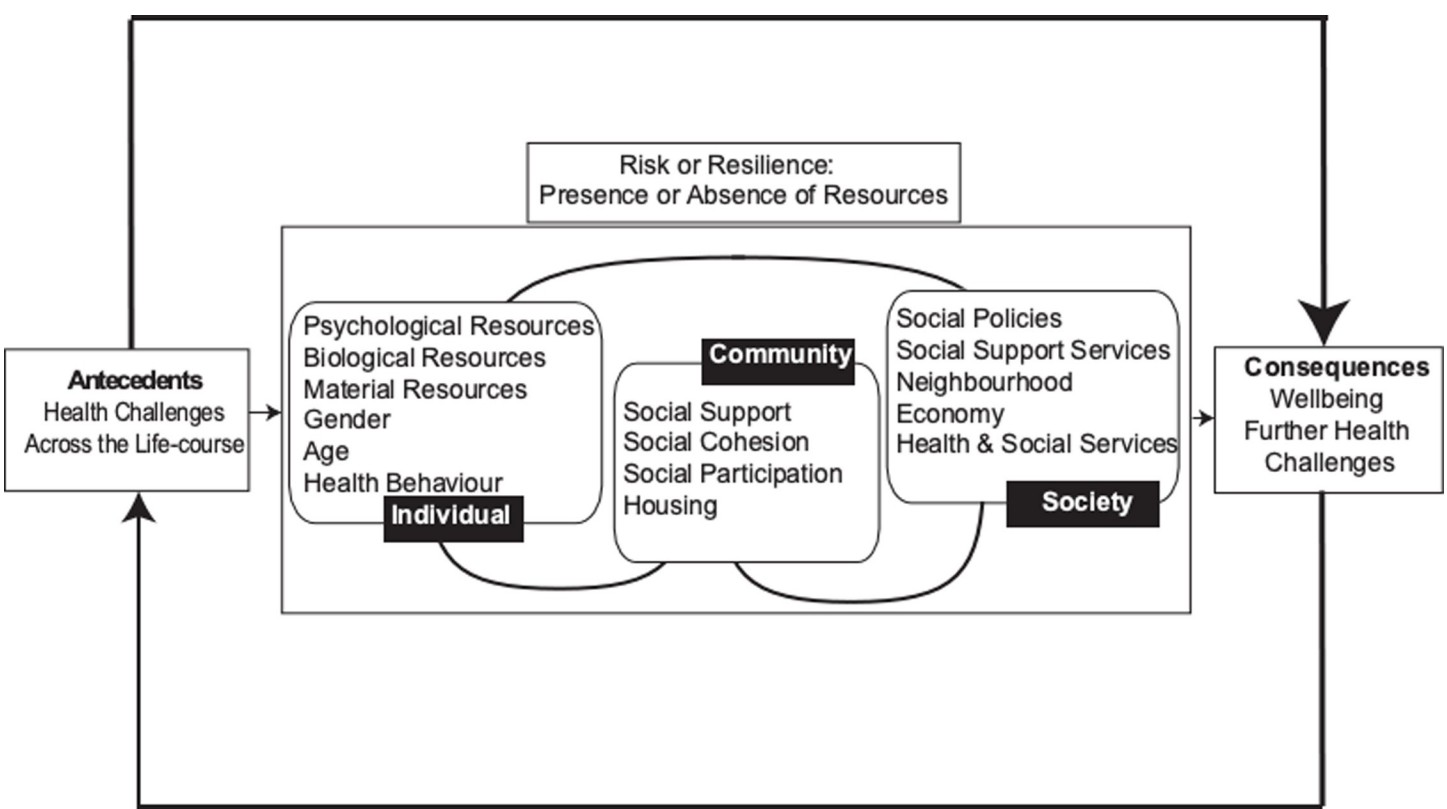

**Fig 1.**

symptoms as measured by the Patient Health Questionnaire-9 item version (PHQ-9) with a score below the cut-off of 10 (PHQ-9 <10) [27] and anxiety symptoms as measured by the Generalized Anxiety Disorder scale with 7 items (GAD-7) with a totale score below 10 (GAD-7 <10) [28]. The remaining participants were defined as Non-Resilient (see [20] for a review of methods to measure resilience). In addition to individual, community, and societal resources we added a COVID-19 specific component which included social distancing, COVID-related anxiety and trauma and exposure to COVID-19. Consistently with findings in the UK by Shevlin and colleagues [5] again about 70% of the sample were resilient, and 64% of the variance was explained by the components of the EMR. Psychological variables such as trait resilience and conscientiousness were associated with Resilient outcomes, whilst loneliness and intolerance of uncertainty were associated with being Non-Resilient. No community variables were found to be significant predictors of Resilient outcomes and region was the only societal variable which was significant, with living in the south associated with Non-resilient outcomes. Among the COVID-specific factors, social distancing was associated with resilience, and COVID-19 anxiety and COVID-19 related trauma were associated with Non-Resilient outcomes. With respect to demographic variables, having children in the household, and higher education level were associated with being Non-Resilient. The study had some limitations. As the Italian survey was not designed to measure resilience in the first instance, some variables, especially with respect to community and society, were not ideal, and others, such as health and welfare service use were not available. Moreover, as this study was cross-sectional, we were unable to determine whether the model was predictive over time [27].

In the current study, we utilise data from the larger C19PRC survey conducted in the UK, which had the opportunity to include additional variables, to further test the EMR's usefulness for explaining resilient outcomes during the pandemic and to compare the findings with the Italian ones. First, we run a replication of the Italian analysis [26], using the same variables, and broadly the same time points in the pandemic (July/August 2020), which was survey Wave 3 in the UK study and equivalent to Wave 1 in the Italian study (July 2020). Both countries had emerged from national lockdowns. Second, we investigate the predictors of resilience one year into the pandemic in the UK (Wave 5; March/April 2021) using baseline data (Wave 1; March/April 2020), and where necessary variables from Wave 3 (July/August 2020) in which we included EMR-specific variables.

Our research questions (RQ) are as follows:

RQ#1: What proportion of the UK sample are resilient, at Wave 3 and Wave 5?

RQ#2: In which ways, if any, does the UK data replicate the Italian Study?

RQ#3: Does the ERM explain resilient outcomes longitudinally?

RQ#4: Which resources predict resilient outcomes and to what extent?

## 2. Materials and methods

This longitudinal study utilises longitudinal panel survey data collected by the C19PRC in the UK, launched in March 2020. In the first year of the pandemic, data was collected through 5 Waves (W1: N = 2025; W2: N = 1406; W3: N = 2019; W4: N = 1796; W5: N = 2520) examining the impact of the pandemic over time. Wave 1 was conducted between March and April 2020, Wave 3 between July and August 2020 and Wave 5 between March and April 2021. This study was approved by the Ethics Committee of the University of Sheffield (reference number: 033759).

### 2.1. Participants

Participants were recruited online, via Qualtrics, with stratified quota sampling to achieve representative sample in terms of age and sex—using 2016 population estimates form Eurostat [29] and household income matched to Office of National Statistics data—for more details see [30]. Qualtrics alerted participants to the study, via Qualtrics. Details of the C19PRC methodology are available elsewhere [6, 31–33]. Participants were eligible to take part if they were at least 18 years old. The online survey took a median completion time of 20 minutes. All respondents were informed about the aims of the study and provided informed written consent.

A cross-sectional and a longitudinal study will be presented. The cross-sectional sample consisted of all respondents to Wave 3 (N = 2019, 51% females, mean age = 45.39, SD = 15.95). The longitudinal study included only those participants who fully completed 1st, 3rd, and 5th Waves (N = 847, 45.04% females, mean age = 51.46, SD = 14.47), thus a subset of the participants of W3. Due to the attrition in longitudinal studies, the longitudinal analysis has a less nationally representative sample, as it underrepresented younger people.

### 2.2. Measures

Below we describe the measures tested in the model; the full list of variables and more details about the survey are available from [6]. For Wave 5 please see McBride et al. [33].

**2.2.1 Defining resilience, the dependent variable.** *Resilience* outcome was defined as a dichotomous variable (1 = Resilient, 0 = Non-Resilient) resulting from the conjoint levels of depressive and anxious symptoms. Resilient individuals had low levels of depressive symptoms (PHQ-9 <10) [27] and low levels of anxiety symptoms (GAD-7 <10) [28]. All other participants were defined as Non-Resilient (PHQ ≥10 or GAD ≥10) [26]. This procedure was followed both for the replication of the Italian study and the longitudinal model.

*Depression* symptoms were measured with the Patient Health Questionnaire-9 (PHQ-9) [27], its 9 items reflect the diagnostic criteria for the major depressive disorder of the Diagnostic and Statistical Manual of Mental Disorders (4th ed., text rev.; DSM-IV-TR; American Psychiatric Association) [34]. Participants indicated how often they had been bothered by each symptom over the past 2 weeks using a four-point Likert type scale from 0 (not at all) to 3 (nearly every day). Total scores range from 0 to 27, with scores of 5–9 = *mild*, 10–14 = *moderate*–used for clinical caseness, and ≥ 15 = *severe* depression. Clinical caseness is defined as ≥ 10. The PHQ-9 psychometric properties are strong [35, 36] and in this study the internal reliability of the scale was excellent (α = .93).

*Anxiety* symptoms were assessed with the Generalized Anxiety Disorder Scale (GAD-7) [28], a 7-item questionnaire measure of the frequency of anxiety symptoms (e.g., trouble relaxing, becoming easily annoyed or irritable). Participants indicated how often they had been affected by each symptom over the past 2 weeks using a four-point Likert scale from 0 (not at all) to 3 (nearly every day). Scores range from 0 to 21 with a cut-off of 10 identifying the generalised anxiety disorder with good sensitivity (.89) and specificity (.82). Clinical caseness is defined as ≥ 10. The GAD-7 showed good psychometric properties [28, 35, 36], and the internal reliability of the scale was excellent (α = .95) in this study. Note that whilst we use the clinical cut-offs for both PHQ- 9 and GAD-7, we are not using them for the purpose of clinical diagnosis.

**2.2.2 Independent variables.** We next describe the independent variables as they appear in the levels of the model in the replication of the Italian study: individual, community, societal and COVID-specific. For the replication, these variables were taken from the C19PRC Wave 3 data, to correspond with the point in time that the Italian data was collected (July 2020). In the longitudinal analyses, they were taken from Wave 1, unless stated otherwise. Variables that appear only in the longitudinal analysis are then described, and the Wave in which they were collected. Binary variables were coded as 1 = present/upper level and 0 = absent/lower level–unless specified (i.e., gender). Scale reliability was evaluated through Cronbach's alpha (α).

*2.2.2.1 Individual resources.* The demographic variables included in the model were: *gender* (women = 1, men = 0; *age* (continuous variable); *education* (split as 0 = until high school and 1 = over high school); *living alone* or not (yes = 1, no = 0); *caring for children in the home* or not (yes = 1, no = 0); *home ownership* (yes = 1, no = 0); *household income above £25,341/year* or not (yes = 1, no = 0); *employed* or not (yes = 1, no = 0).

In addition, we included *precarious health*. The respondents were asked: *"Were you diagnosed with a health condition (e.g., heart or lung disease; diabetes; cancer) before December 31st, 2019 (i.e. before the start of the coronavirus COVID-19 outbreak)?".* Poor health was coded as 1 (present) or 0 (absent).

The following **psychological variables** were also included:

*Death anxiety* (DA) was measured using the Death Anxiety Inventory (DAI) [37], which has 17 items across 4 dimensions. *DAI Acceptance* (6 items, α = .87) concerns the acceptance of the individual emotional dimension about death and its meaning—an item example is "I find it really difficult to accept that I have to die". *DAI Externally Generated* (4 items, α = .78) refers to situations or elements with an external reference to death in our cultural context (e.g., coffins, cemeteries) that can trigger unpleasant feelings—an item example is "I get upset when I am in a

cemetery". *DAI Finality* (4 items, α = .90) concerns a spiritual dimension of concerns about mortality and the limits of human existence—an item example is "The idea that there is nothing after death frightens me". *DAI Thoughts* (3 items, α = .82) is related to the frequent thoughts and concerns about own death—an item example is "I frequently think of my own death". The response format is a 5-point Likert type scale from 1 (totally disagree) to 5 (totally agree).

*Personality traits* were assessed with the Big Five Inventory (BFI) [38], a 10-item measure with 5 scales, each for a personality trait: *Extraversion* (e.g., *"I see myself as someone who is outgoing, sociable"*), *Agreeableness* (e.g., *"I see myself as someone who is generally trusting"*), *Consciousness* (e.g., *"I see myself as someone who does a thorough job"*), *Neuroticism* (e.g., *"I see myself as someone who gets nervous easily"*), *Openness to Experience* (e.g., *"I see myself as someone who has an active imagination"*). Items are rated on a 5-step scale from 1 (disagree strongly) to 5 (agree strongly), each scale ranges from 2 to 10. The BFI-10 has good reliability and validity [38].

*Loneliness* was measured with the Loneliness Scale (LS) [39], a 3-item tool assessing the frequency of social (dis)connectedness (e.g., lacking companionship, isolation from others) in large-scale population surveys. An item example is "How often do you feel isolated from others?". The response format is a 3-point scale from 1 (hardly ever) to 3 (often) with a minimum of 3 and a maximum of 9. In this study, α was .89.

*Self-esteem* was assessed with the Single-Item Self-esteem Scale (SISES) [40], respondents rated their agreement with one statement ("*I have high self-esteem*") using a 7-point Likert scale from 1 (not very true of me) to 7 (very true of me).

*Trait resilience* was measured with the Brief Resilience Scale (BRS) [41] with 6 items (e.g., *"I tend to bounce back quickly after hard times"*) scored on a 5-point Likert scale from 1 "strongly disagree" to 5 "strongly agree", total scores range from 5 to 30. The BRS has good psychometric properties [41]. In this study, α was .88.

**2.2.3 Community variables.** *Social Support* was measured in its *Emotional* and *Instrumental* components with the Modified Medical Outcome Social Support Survey (MOS-SS) [42], asking participants: *"People sometimes look to others for companionship, assistance, or other types of support. How often are each of the following kinds of support available to YOU if you need it?".* Each component was assessed with 4 items scored on a 5-point Likert-type scale from 1 (None of the time) to 5 (All of the time). Higher scores are associated with higher perceived support. The reliability was excellent for the *Emotional* (α = .93). and *Instrumental* components (α = .95)

**2.2.4 Societal resources.** *Religious*. Participants were asked whether they identified with any religion (Catholic, Jewish, Muslim, other = 1), or not (atheist, agnostic, none = 0). Whilst this variable could also be seen as an individual resource, it was included as a societal resource because of its broader cultural significance [25, 43].

**2.2.5 COVID-19 specific variables.** We included several variables specifically related to COVID-19.

*COVID-19 anxiety* was measured with a single item *("How anxious are you about the coronavirus COVID-19 pandemic?")* on an electronic visual analogue scale to indicate the degree of anxiety from 0 "not at all anxious" on the left to 100 "extremely anxious" with 10-point increments. Higher scores reflected higher levels of COVID-19 related anxiety.

*Post-traumatic stress disorder symptoms in relation to the COVID-19* experience (*PTSD Symptoms*) was measured with the International Trauma Questionnaire (ITQ) [44] referring to the last month with the following instructions: *"You will be asked questions about different ways that people sometimes react following a traumatic or stressful life event. Please answer the following questions in relation to your experience of the COVID-19 pandemic. Please read each item carefully, then select one of the answers to indicate how much you have been bothered by*

*that problem in the past month"*. The ITQ has 6 items encompassing three clusters of symptoms of Re-experiencing, Avoidance, and Sense of Threat. A 5-point Likert scale from 0 (Not at all) to 4 (Extremely) generates scores ranging from 0 to 24. A was .93 in this study.

*Exposure to COVID-19* was defined as the experience of self, family or acquaintance being infected or tested for COVID-19 (whether the outcome was positive or negative: as there may be anticipatory anxiety), or a family member or acquaintance having died because of COVID-19, as well having been in self-isolation because of the (suspected) infection. Those exposed to COVID-19 were scored 1 and those not exposed were scored 0.

The following variables were included in the longitudinal analyses only.

*Intolerance of Uncertainty (Wave 1)* was assessed with the Intolerance of Uncertainty Scale Short Form (IUS-12) [45], with 12 items scored on a 5-point Likert scale from 1 "*not at all characteristics of me"* to 5 "*entirely characteristic of me"*. It includes both prospective cognitions "*Unexpected events are negative and should be avoided"*) and inhibitory cognitions "*Uncertainty leads to the inability to act"*) about uncertainty that are well described in a single dimension [45]. The total score ranges from 12 to 60. The IUS-12 has good psychometric properties [45], and α was .90 in this study.

*Trust in People (Wave 1)* was assessed with the following *ad-hoc* question "*Generally speaking, would you say that most people can be trusted or that you need to be very careful in dealing with people?"* with a response format scored on a 5-point Likert-type scale ranging from 1 (= "Mostly people can be trusted") to 5 (= "Need to be very careful"). Scores were dichotomized according to the median that was 3, resulting with 1110 persons saying that they did trust people and 778 who did not.

*Neighbourhood Connectedness (Wave 1)* was assessed with three measures. Belongingness to the neighbourhood was measured with a single question "*How strongly do you feel you belong to your immediate neighbourhood?"* scored on a 4-point response format from 1 "not at all" to 4 "very strongly". There were two specific questions on trust in neighbours asking about willingness to leave house keys to them, and whether they could be asked to buy groceries in case of need, scored on a 4-point scale from 1 (= "very uncomfortable") to 4 (= "very comfortable"). Total scores range from 2 to 8, with higher scores associated with higher trust and connectedness.

*Belongingness to Wider Neighbourhood* (Wave 1) was assessed with 2 questions "*How much do you identify with (feel a part of, feel love toward, have concern for) your community?"* and "*How much would you say you feel involved when bad things happen to your community?"*– scored on a 5-point format from 1 (= "not at all") to 5 (= "a lot"). Scores were dichotomized according to the median (= 4).

*Hygiene Behaviours* (Wave 1) were measured with 17 items based on the Capability, Opportunity, Motivation-Behaviour, version 1 (COM-B) [46, 47]. Participants were asked "*Please answer the following questions to indicate the extent to which the following statements are true for you with respect to maintaining hygienic practices (e.g., hand washing frequently, cleansing surfaces) during the COVID-19 pandemic"*. Answers were scored on a 5-point Likert scale ranging from 1 "strongly agree" to 5 "strongly disagree". In this study, α was .84.

*Social Distance Behaviours* (Wave 1) were assessed with 17 statements according to the COM-B [46, 47]. Participants were asked "*Please answer the following questions to indicate the extent to which the following statements are true for you with respect to social distancing (e.g., avoiding crowds, maintaining personal distance, avoiding non-essential meetings, less socialising in public) during the COVID-19 pandemic"*. Answers were scored on a 5-point Likert scale from 1 "strongly agree" to 5 "strongly disagree". A was .86 in this study.

*Service Use* was assessed at Wave 5 by asking participants to "*. . .answer the following questions in relation to your attendance at health appointments and use of healthcare services since*

*the beginning of the pandemic. Since the pandemic, have you attended. . .*" any of the 13 following health services: dentist; sight test; hearing test; other health check-ups; physiotherapist; occupational therapist; chiropodist; social worker; speech therapist; day centre; outpatient services; inpatient services; and General Practitioner appointments. Each service was assessed by one dedicated item. Items were scored as '*Yes, I had a consultation in person*' (= 1), '*Yes, I had a consultation online or by telephone*' (= 2), '*No, the pandemic prevented me from doing so*' (= 3), '*No, I didn't need this service during the pandemic*' (= 4). Since we were interested in access to services during the pandemic, the response options were recoded as follows. Those who were prevented from using at least one service because of the pandemic were scored 0, and those who used the services or did not need them were scored as 1.

*Social Engagement* (Wave 5) was measured by asking participants "*The following items are about your social contact with family and friends. Please read each item and indicate which option best applies to you*". The 5 dichotomous items (yes = 1, no = 0) concerned the social interactions of the participants through social media, personal calls, letters/emails, meetings and friends living in the surroundings. Summing the answers, the score ranged from 0 to 5 with higher scores indicating higher social engagement. The scale was based on the Brief Assessment of Social Engagement Scale (BASE) [48] which measured both actual and symbolic social engagement, updated to the current time. α was .47.

*Self-Rated Health* (Wave 5) was measured through one question–- "*Compared to someone your own age, would you say your health has on the whole been. . .*" and asking participants to move a slider on a scale from poor (= 1) to excellent (= 5), with higher scores indicating higher levels of perceived self-rated health.

## 2.3. Statistical analysis

Descriptive analyses were used to describe the sample and the psychosocial characteristics of participants. The analytical plan was pre-registered on Open Science Framework (https://osf.io/8dn27). Within the family of generalized linear models, a binary logistic univariate multiple regression was fitted to predict the *Resilient* (= 1) or *Non-Resilient* (= 0) outcome of participants relying on multiple predictor variables that were progressively entered in blocks according to the EMR model. Regressions allowed to estimate the effect of each predictor variable on the dependent variable once the effect of the other variables has been considered; in this model the interactions among variables were not considered.

Model fit [49] was evaluated through the Akaike Information Criteria (AIC), Bayesian Information Criteria (BIC), Pseudo $R^2$ of Cragg-Uhler [50], Odds ratio (OR) and the 95% confidence intervals (CI) were calculated and reported.

The first set of regression models is based on cross-sectional data of Wave 3 from the UK study (N = 2019) (6) and focuses on predicting state resilience at W3 (resilient = 1 or not resilient = 0) with multiple independent variables progressively entered in blocks according to the EMR–thus replicating and going beyond the previous Italian study [26].

The second set of regression models is based on longitudinal data from the UK study. It aims to predict resilience at Wave 5 (resilient = 1 or not resilient = 0) [33] still relying on multiple predictors variables that were measured over time in the same individuals at Wave 1, Wave 3, and Wave 5 (N = 847). As in the previous model, the predictors were entered in blocks in line with the EMR [21].

# 3. Results

## 3.1. Cross-sectional replication of the Italian study

**3.1.1 Characteristics of sample.** Table 1 shows the characteristics of the sample at Wave 3 [31] including a total of 2019 respondents with a mean age of 45.39 years (SD = 15.95) and a representative gender-balance, 51% were women. Most of participants had qualifications equal to or above high school (e.g., degree) (61%), whilst the remaining 49% attained a lower level of education (O-Level/GCSE or similar). Only 21% of the sample lived alone.

**3.1.2 Replication of the Italian study: Regression.** The cross-sectional model of binary logistic regression had as dependent variable state Resilience at Wave 3 (1 = *Resilient*, 0 = *Non-Resilient*) and all the predictor variables were measured at Wave 3. Predictors were consecutively entered in blocks according to the EMR [21]. Table 2 shows the results of the cross-sectional regression analysis that was performed on 848 complete cases who completed 1st, 3rd, 5th Waves.

**Table 1. Characteristics of the sample at Wave 3.**

| Characteristic | n (%) N = 2,019 |
|---|---|
| W3 Resilient | 1,446 (72%) |
| W3 Gender: women | 1,020 (51%) |
| W3 Education ≥ high school | 1,227 (61%) |
| W3 Living alone | 414 (21%) |
| W3 Caring for children in household | 532 (56%) |
| W3 House property/ownership | 1,478 (73%) |
| W3 Income above £25,341/year (y) | 1,267 (63%) |
| W3 Employed (y) | 1,238 (61%) |
| W3 Precarious Health (y) | 1,037 (51%) |
| Variable | Mean (SD) N = 2,019 |
| W3 Age | 45.39 (15.95) |
| W3 DA Acceptance | 14.62 (5.51) |
| W3 DA Externally Generated | 10.62 (3.84) |
| W3 DA Finality | 10.22 (4.43) |
| W3 DA Thoughts | 7.04 (3.11) |
| W3 Extraversion | 5.62 (2.00) |
| W3 Agreeableness | 6.59 (1.66) |
| W3 Conscientious | 7.20 (1.76) |
| W3 Neuroticism | 6.09 (2.12) |
| W3 Openness | 6.65 (1.59) |
| W3 Loneliness | 5.08 (1.93) |
| W3 Self Esteem | 3.77 (1.63) |
| W3 Trait Resilience | 18.81 (5.17) |
| W3 Social Support: Emotional | 13.46 (4.93) |
| W3 Social Support: Instrumental | 13.12 (5.41) |
| W3 COVID-19 anxiety | 50.87 (27.66) |
| W3 PTSD symptoms | 4.26 (5.60) |

Note: W3 = Wave 3; y = yes; DA = Death Anxiety; PTSD = post-traumatic stress disorder.

Regarding the first research question, 72% (n = 1446) of participants at W3 showed resilient outcomes–this result is in line with the Italian study [26].

**Table 2. Cross-sectional regression analysis at Wave 3 in the United Kingdom (N = 2,019).**

| Characteristic | Block 1 | | | Block 2 | | | Block 3 | | | Block 4 | | | Block 5 | | |
|---|---|---|---|---|---|---|---|---|---|---|---|---|---|---|---|
| | OR | 95% CI | p | OR | 95% CI | p | OR | 95% CI | p | OR | 95% CI | p | OR | 95% CI | p |
| W3 Gender (women) | 0.85 | 0.69, 1.05 | .142 | 1.16 | 0.76, 1.78 | .492 | 1.20 | 0.77, 1.89 | .421 | 1.21 | 0.77, 1.90 | .405 | 1.12 | 0.67, 1.86 | .667 |
| **W3 Age year** | **1.04** | **1.03, 1.05** | **< .001** | **1.03** | **1.01, 1.05** | **< .001** | **1.04** | **1.02, 1.05** | **< .001** | **1.04** | **1.02, 1.06** | **< .001** | **1.02** | **1.00, 1.04** | **.038** |
| W3 Education ≥ h.s. | 0.97 | 0.78, 1.21 | .798 | 0.98 | 0.64, 1.48 | .906 | 1.00 | 0.64, 1.54 | .982 | 1.01 | 0.65, 1.56 | .976 | 1.08 | 0.67, 1.75 | .753 |
| W3 Living alone | 0.88 | 0.46, 1.69 | .693 | 0.34 | 0.04, 3.07 | .336 | 0.27 | 0.03, 2.68 | .263 | 0.33 | 0.03, 3.35 | .351 | 0.83 | 0.06, 11.3 | .890 |
| W3 Children in house | 1.00 | 0.93, 1.07 | .941 | 0.90 | 0.72, 1.12 | .356 | 0.87 | 0.69, 1.09 | .227 | 0.89 | 0.71, 1.12 | .317 | 0.98 | 0.75, 1.27 | .861 |
| W3 House property | 1.13 | 0.90, 1.43 | .300 | 1.29 | 0.84, 1.96 | .243 | 1.35 | 0.86, 2.11 | .191 | 1.35 | 0.87, 2.12 | .185 | 1.36 | 0.83, 2.23 | .228 |
| W3 Income > £25,341/y | **1.66** | **1.32, 2.09** | **< .001** | 1.22 | 0.79, 1.88 | .364 | 1.28 | 0.81, 2.00 | .287 | 1.24 | 0.79, 1.95 | .340 | 1.50 | 0.90, 2.49 | .118 |
| W3 Employment (y) | 1.17 | 0.94, 1.47 | .168 | 0.77 | 0.50, 1.19 | .241 | 0.72 | 0.45, 1.13 | .157 | 0.72 | 0.45, 1.14 | .161 | 0.91 | 0.54, 1.51 | .714 |
| W3 Precarious Health (y) | 1.08 | 0.87, 1.33 | .486 | **0.46** | **0.27, 0.79** | **.005** | **0.50** | **0.28, 0.87** | **.015** | **0.50** | **0.28, 0.88** | **.016** | 0.54 | 0.28, 1.02 | .058 |
| **Characteristic** | | | | Block 2 | | | Block 3 | | | Block 4 | | | Block 5 | | |
| | | | | OR | 95% CI | p | OR | 95% CI | p | OR | 95% CI | p | OR | 95% CI | p |
| W3 DA Acceptance | - | - | - | **0.92** | **0.86, 0.99** | **.030** | **0.92** | **0.85, 1.00** | **.043** | 0.93 | 0.86, 1.00 | .050 | 0.96 | 0.88, 1.05 | .374 |
| W3 DA External | - | - | - | 1.06 | 0.99, 1.14 | .109 | 1.07 | 0.99, 1.16 | .086 | 1.07 | 0.99, 1.16 | .080 | 1.09 | 1.00, 1.18 | .058 |
| W3 DA Finality | - | - | - | **1.11** | **1.02, 1.21** | **.021** | **1.11** | **1.01, 1.22** | **.037** | **1.11** | **1.01, 1.22** | **.036** | 1.07 | 0.96, 1.18 | .232 |
| **W3 DA Thoughts** | - | - | - | **0.79** | **0.72, 0.86** | **< .001** | **0.78** | **0.71, 0.86** | **< .001** | **0.78** | **0.71, 0.86** | **< .001** | **0.84** | **0.75, 0.94** | **.002** |
| **W3 Extraversion** | - | - | - | **0.76** | **0.67, 0.86** | **< .001** | **0.76** | **0.66, 0.86** | **< .001** | **0.76** | **0.66, 0.86** | **< .001** | **0.82** | **0.71, 0.94** | **.005** |
| W3 Agreeableness | - | - | - | 1.02 | 0.90, 1.17 | .714 | 1.03 | 0.90, 1.18 | .668 | 1.04 | 0.90, 1.19 | .614 | 1.04 | 0.89, 1.21 | .619 |
| W3 Conscientious | - | - | - | 1.04 | 0.91, 1.18 | .561 | 1.02 | 0.89, 1.16 | .810 | 1.01 | 0.88, 1.15 | .883 | 1.01 | 0.87, 1.17 | .885 |
| **W3 Neuroticism** | - | - | - | **0.69** | **0.59, 0.80** | **< .001** | **0.70** | **0.59, 0.81** | **< .001** | **0.69** | **0.59, 0.80** | **< .001** | **0.67** | **0.56, 0.79** | **< .001** |
| W3 Openness | - | - | - | 0.95 | 0.83, 1.08 | .428 | 0.96 | 0.84, 1.10 | .548 | 0.95 | 0.82, 1.09 | .433 | 0.96 | 0.83, 1.12 | .619 |
| **W3 Loneliness** | - | - | - | **0.63** | **0.56, 0.71** | **< .001** | **0.60** | **0.53, 0.69** | **< .001** | **0.61** | **0.53, 0.69** | **< .001** | **0.69** | **0.60, 0.79** | **< .001** |
| **W3 Self-Esteem** | - | - | - | **1.19** | **1.03, 1.38** | **.018** | **1.22** | **1.05, 1.43** | **.010** | **1.24** | **1.06, 1.44** | **.007** | **1.34** | **1.13, 1.60** | **.001** |
| W3 Trait Resilience | - | - | - | 1.06 | 0.99, 1.12 | .078 | 1.05 | 0.99, 1.12 | .101 | 1.06 | 0.99, 1.12 | .094 | 1.01 | 0.94, 1.08 | .806 |
| **Characteristic** | | | | | | | Block 3 | | | Block 4 | | | Block 5 | | |
| | | | | | | | OR | 95% CI | p | OR | 95% CI | p | OR | 95% CI | p |
| W3 Soc Supp Emotional | - | - | - | - | - | - | 0.97 | 0.90, 1.04 | .377 | 0.96 | 0.89, 1.03 | .298 | 0.95 | 0.88, 1.03 | .245 |
| W3 Soc Supp Instrument. | - | - | - | - | - | - | 1.05 | 0.98, 1.12 | .147 | 1.05 | 0.99, 1.13 | .126 | 1.07 | 0.99, 1.16 | .079 |
| W3 Religious (yes) | - | - | - | - | - | - | - | - | - | 0.71 | 0.46, 1.11 | .134 | 0.96 | 0.58, 1.57 | .868 |
| W3 C19 anxiety | - | - | - | - | - | - | - | - | - | - | - | - | 1.00 | 0.99, 1.01 | .463 |
| **W3 PTSD symptoms** | - | - | - | - | - | - | - | - | - | - | - | - | **0.79** | **0.75, 0.83** | **< .001** |
| W3 C19 Exposure | - | - | - | - | - | - | - | - | - | - | - | - | 0.77 | 0.47, 1.27 | .304 |

Note: The color of the lines inform about the effect of each predictor on state resilience. White lines mean that the predictor is not statistically significant; green lines mean that a predictor has a statistically significant positive effect on resilience; and red lines mean that the predictor has a statistically significant negative effect on resilience.

OR: odds ratio; 95% CI = confidence interval at 95%; p = p-value; W3 = Wave 3; h.s. = high school; y = yes; DA = death anxiety; Soc. Supp. = Social support; PTSD = post traumatic stress disorder; C19 = COVID-19.

*Block 1* included the individual demographic variables, and the model showed a good fit to the data ($\chi^2_{(9)}$ = 105.625, p < .001; AIC = 1032.219, BIC = 1079.648), explaining 16% of variance of the dependent variable (Pseudo-$R^2$ = 0.160). The predictors significantly and positively associated with resilience were higher *age* (OR = 1.05, CI = 1.03–1.06, p < .001) and *income* above £25,341/y (OR = 1.55, CI = 1.11–2.16, p < .010) whilst *precarious health* had a significant negative association with *Resilient* outcomes (OR = 0.38, CI = 0.25–0.58, p < .001).

*Block 2* added the individual psychological variables. The model had good fit ($\chi^2_{(21)}$ = 434.996, p < .001; AIC = 726.848, BIC = 831.191) and explained 55% of variance (Pseudo-$R^2$ =

0.548), adding an incremental .532 $R^2$. Among the demographic variables, *age* (OR = 1.03, CI = 1.01–1.05, p < .001) and *precarious health* (OR = 0.46, CI = 0.27–0.79, p = .005) were still significantly associated with *Resilient* outcomes, respectively with a positive and a negative direction. Among the psychological variables, both *DAI-Finality* (OR = 1.11, CI = 1.02–1.21, p = .021) and *Self-Esteem* (OR = 1.19, CI = 1.03–1.38, p = .018) positively associated with *Resilient* outcomes, whilst the variables with a significant negative association with *Resilient* outcomes were *DAI Acceptance* (OR = 0.92, CI = 0.86–0.99, p = .030), *DAI Thoughts* (OR = 0.79, CI = 0.72–0.86, p < .001), *Extraversion* (OR = 0.76, CI = 0.67–0.86, p < .001), *Neuroticism* (OR = 0.69, CI = 0.59–0.80, p < .001), and *loneliness* (OR = 0.63, CI = 0.56–0.71, p < .001).

*Block 3* added to the model the community variables that are represented by both *Instrumental Social Support* and *Emotional Social Support*. With a good fit ($\chi^2_{(23)}$ = 431.873, p < .001; AIC = 676.588, BIC = 789.049) the model explained 57% of variance (Pseudo-$R^2$ = 0.568), but added only .02 of $R^2$. The *Social Support* variables were not significantly associated with *Resilient* outcomes and the other significant predictors were the same as in the previous block.

In Block 4 we added the only societal variable, namely being *religious* or not. The model provided good fit ($\chi^2_{(24)}$ = 434.144, p < .001; AIC = 676.317, BIC = 793.463) and still explained the 57% of variance (Pseudo-$R^2$ = 0.570) but with an incremental $R^2$ of .002. Indeed, being *religious* or not was not associated with Resilience.

Lastly, in Block 5 the COVID19-related variables were added, the model showed a good fit ($\chi^2_{(27)}$ = 540.469, p < .001; AIC = 575.992, BIC = 707.196) and explained the 70% of variance (Pseudo-$R^2$ = 0.669) with an incremental $R^2$ of .099 compared to the previous block. *Age* was the only significant predictor among the demographics (OR = 1.02, CI = 1.00–1.04, p = .038). Among the psychological factors, only *Self-Esteem* (OR = 1.34, CI = 1.13–1.60, p = .001) was significantly positively associated with resilience whilst the significant predictors negatively associated with resilience were *Death Anxiety Thoughts* (0.84, CI = 0.75–0.94, p = .002), *Extraversion* (OR = 0.82, CI = 0.71–0.94, p = .005), *Neuroticism* (OR = 0.67, CI = 0.56–0.79, p < .001), and *Loneliness* (OR = 0.69, CI = 0.60–0.79, p < .001). Concerning the COVID-19-related variables, the severity of *PTSD Symptoms* related to COVID-19 was the only significant predictor negatively associated with *Resilience* (OR = 0.79, CI = 0.75–0.83, p < .001).

Compared to the Italian cross-sectional model [26], the UK model shows better fit in explaining the data according to the ERM (UK model: $\chi^2_{(27)}$ = 540.469, p = < .001; AIC = 575.992, BIC = 707.196, Pseudo-$R^2$ = 0.669; IT model: Nagelkerke $R^2$ = 0.642; AIC = 707.3; BIC = 885.2; Residual deviance: 635.28, df = 998).

Fig 2 shows the odds ratio plot of all the factors in the cross-sectional UK model at Wave 3.

Fig 3 shows the comparison of the odds ratio plot of all factors between Italy and UK.

Fig 4 shows the odds ratio plot only of the factors significantly associated with resilience in the cross-sectional UK model (W3).

## 3.2. Longitudinal model

**3.2.1 Characteristics of sample.** The longitudinal model was fitted on a sample of 847 complete cases who completed the surveys at W1, W3, and W5. This sample had a mean age of 51.46 (SD = 14.47) and was balanced in terms of gender (women = 381, 45.04%).

Considering resilience at Wave 5, 80% of the sample showed *Resilient* outcomes defined as the absence of severe anxiety and depression. The proportions of UK participants who were resilient showed a statistically significant increase from Wave 3 (1446 of 2019, 72%) to Wave 5 (677 of 847, 80%) as found by the one-sided (less) two-sample test for equality of proportions

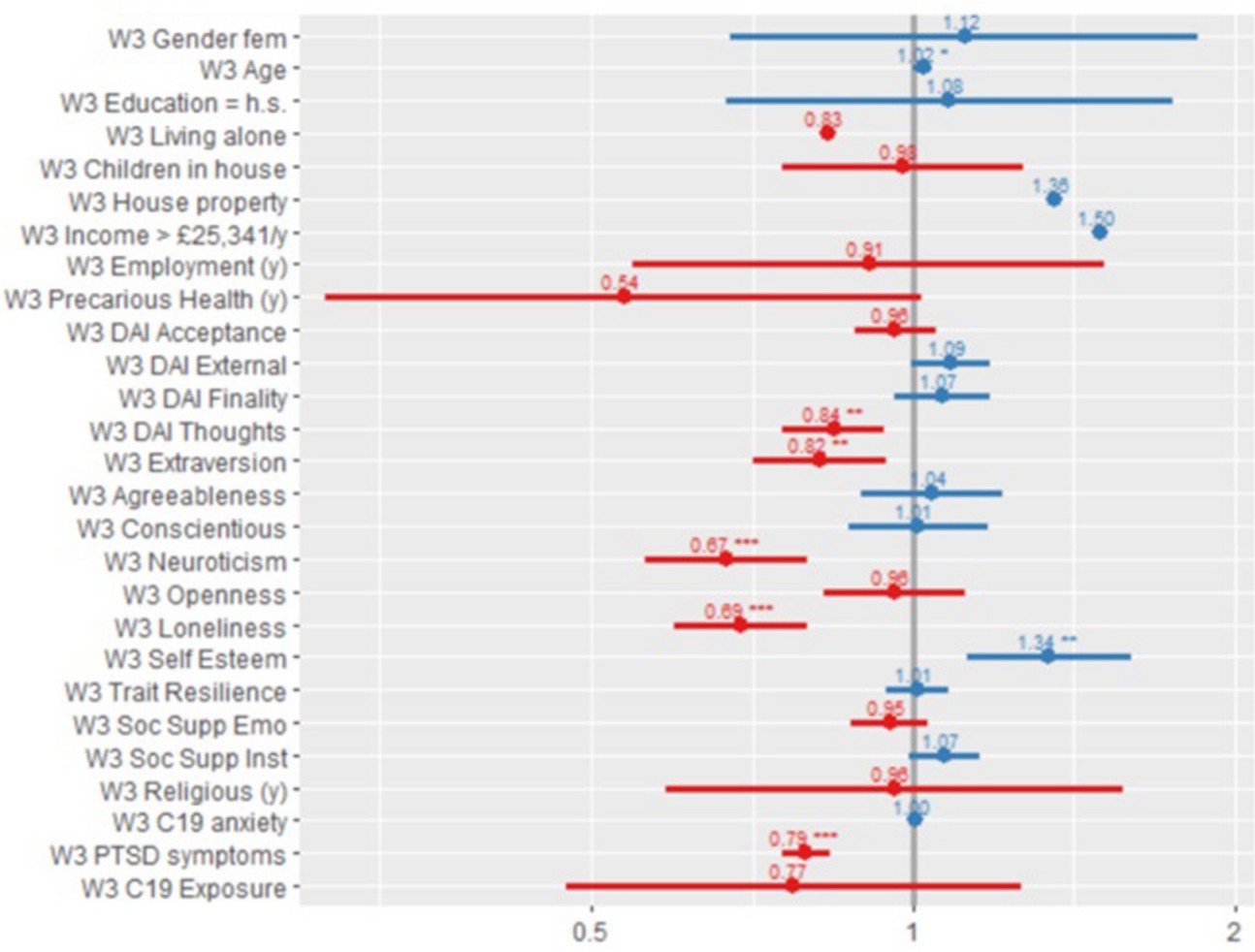

**Fig 2.**

with continuity correction ($X^2$ = 21.024, df = 1, p < .001, 95% CI = -1–0.054). Table 3 reports the complete descriptive statistics of the variables included in the longitudinal UK model.

**3.2.2 Longitudinal model: Regression.** The longitudinal model included 847 participants. The variables were added in subsequent blocks according to the EMR [21]. Table 4 shows the results of the longitudinal regression model.

*Block 1* included the individual demographic variables. The model showed good fit indexes ($\chi^2_{(9)}$ = 44.057, p < .001; AIC = 824.838, BIC = 872.243) and explained the 8% of variance (Pseudo-$R^2$ = 0.080). The variables significantly associated with *Resilient* outcomes were *male gender* (OR = 1.30, p = .046), higher *age* (OR = 1.03, CI = 1.02–1.04, p < .001), *income above £25,341/Y* (OR = 1.54, CI = 1.04, 2.29, p = .032), and *being employed* (OR = 2.75, CI = 1.12–6.69, p = .025). Conversely, *precarious health* was significantly associated with a *Non-Resilient* outcome (OR = 0.62, CI = 0.42, 0.94, p = .023).

*Block 2* added the individual psychological variables, showing good fit ($\chi^2_{(23)}$ = 246.007, p < .001; AIC = 639.001, BIC = 752.545) and explained the 40% of variance (Pseudo-$R^2$ = 0.403), with an $R^2$ increment of 32% compared to the previous block. Regarding the significant predictors, none of the first block remained significant whilst among the psychological variables *self-rated health* (W5) (OR = 1.87, CI = 1.47, 2.39, p < .001) was significantly

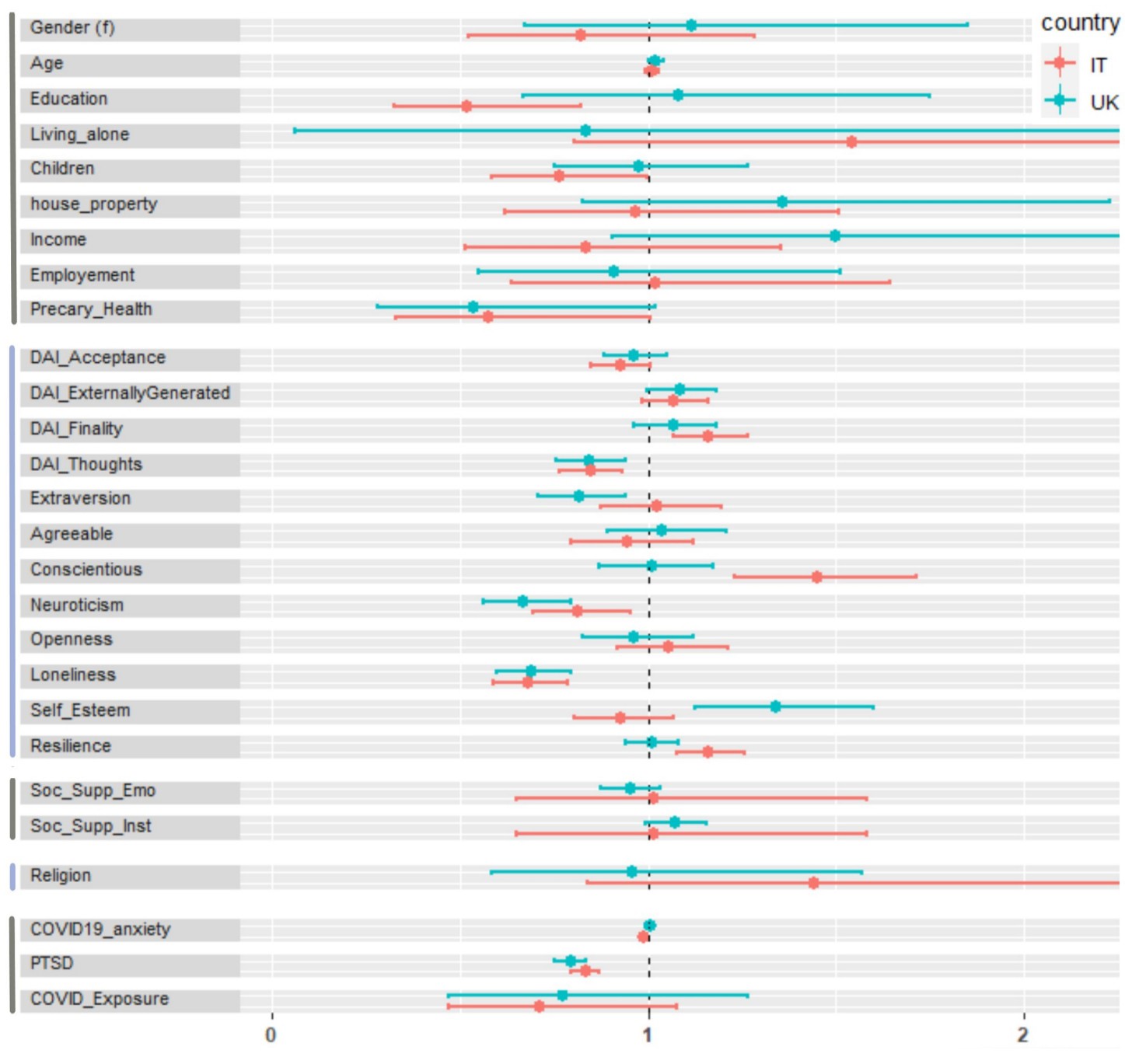

**Fig 3.**

associated with *Resilient* outcomes (OR = 1.87, p < .001) and the predictors associated with *Non-Resilient* outcomes were *DAI Thoughts* (W1) (OR = 0.91, CI = 0.83–1.00, p = .043), *Neuroticism* (W1) (OR = 0.85, CI = 0.73–0.99, p = .037), and *loneliness* (W1) (OR = 0.76, CI = 0.67–0.86, p < .001).

In *Block 3* the community variables were added, the model had good fit ($\chi^2_{(30)}$ = 242.950, p < .001; AIC = 614.557, BIC = 759.819) and explained the 42% of variance (Pseudo-$R^2$ =

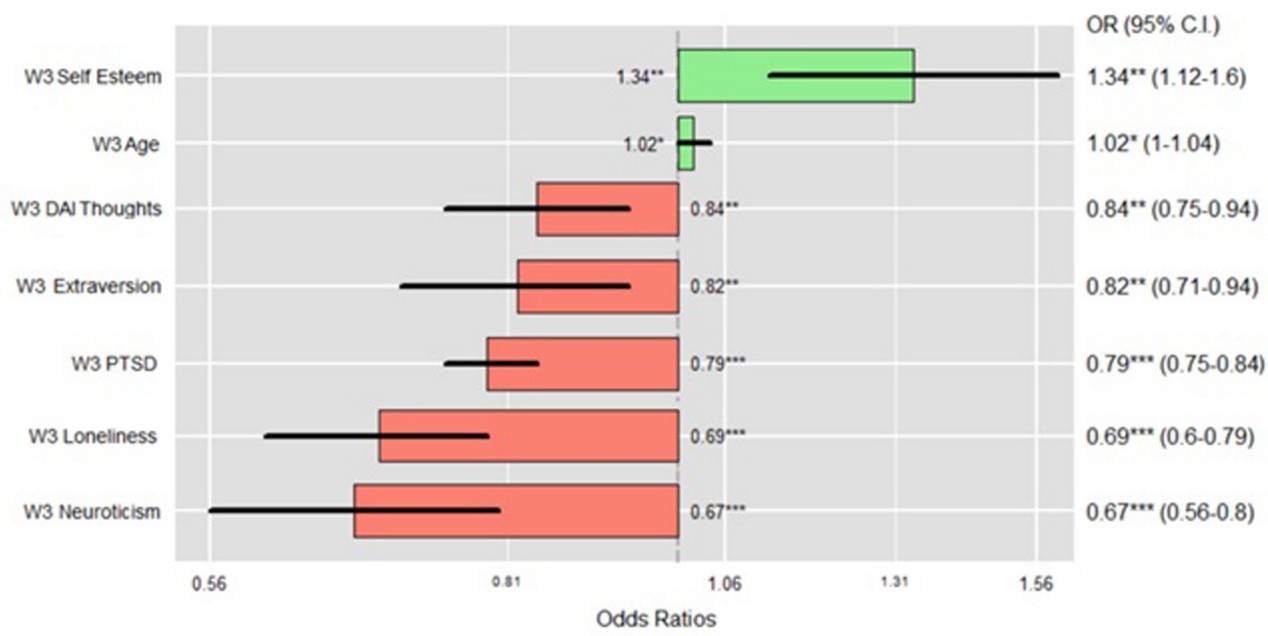

**Fig 4.**

0.416), adding less than 2% compared to the previous block. There were no significant demographic predictors of *resilient* outcomes at W5. Among the psychological variables, *self-rated health* (W5) (OR = 1.75, CI = 1.36–2.25, p < .001) was still significantly associated with *Resilient* outcomes at W5 as well as *Neuroticism* (OR = 0.82, CI = 0.70–0.96, p = .016) and *loneliness* (OR = 0.75, CI = 0.66–0.85, p < .001) that were still negatively associated with *Resilient outcomes*. The *DAI Thoughts* was no longer significant (OR = 0.92, CI = 0.84, 1.01, p = .097). Among the newly added community variables, both *belongingness to neighbourhood* (W1) (OR = 1.67, CI = 1.03–2.72, p = .038) and *Instrumental Social Support* (W3) (OR = 1.08, CI = 1.01–1.16, p = .036) were significantly associated with *Resilient* outcomes.

*Block 4* added the predictors about society, namely being *religious* or not (W1) with good fit ($\chi^2_{(31)}$ = 245.552, p < .001; AIC = 613.955, BIC = 763.902), but the explained variance was almost the same as in the previous model and the delta increase in the Pseudo-$R^2$ was negligible (Pseudo-$R^2$ = 0.419; $\Delta$ Pseudo-$R^2$ = +0.003). Indeed, *religion* was not significantly associated with *Resilient* outcomes (OR = 0.68, CI = 0.42–1.09, p = .110). The other significant predictors were the same as in the previous model: *self-rated health* (OR = 1.77, CI = 1.38–2.28, p < .001); *Neuroticism* (OR = 0.82, CI = 0.70–0.97, p = .018); *loneliness* (OR = 0.75, CI = 0.66, 0.86, p < .001); *neighbourhood belongingness* (OR = 1.69, CI = 1.04–2.76, p = .035); *Instrumental Social Support* (W3) (OR = 1.08, CI = 1.00–1.16, p = .044).

Lastly, *Block 5* added the COVID-19-related variables and showed a good fit ($\chi^2_{(37)}$ = 283.802, p < .001; AIC = 587.706, BIC = 765.768) with an explained variance equal to 47% (Pseudo-$R^2$ = 0.474) and an increment of .055 in $R^2$ compared to the previous model. There were no significant demographic predictors. Among the psychological variables, the predictors significantly associated with *Resilient* outcomes were *self-rated health* (OR = 1.89, CI = 1.44–2.50, p < .001), *DAI Externally Generated* (OR = 1.09, CI = 1.00–1.19, p = .047), *Openness* (OR = 1.17, CI = 1.02–1.34, p = .028), *Loneliness* (OR = 0.79, CI = 0.68–0.90, p < .001), and *self-esteem* (OR = 1.30, CI = 1.08–1.58, p = .007). Among the community block, *neighbourhood belongingness* (OR = 1.96, CI = 1.18–3.30; p = .010) was significantly and positively associated

**Table 3. Descriptive statistics of the variables included in the longitudinal model.**

| Variable | n (%) / Mean (SD); N = 847 |
|---|---|
| W5 Resilient or not | 677 (79.93%) |
| W1 Gender: Women | 381 (45.04%) |
| W1 Age | 51.46 (14.47) |
| W1 Education ≥ High school | 538 (63.52%) |
| W1 Living alone (y) | 192 (22.67%) |
| W1 Children in household (y) | 178 (21.02%) |
| W1 House property (y) | 673 (79.46%) |
| W1 Income > £25,341/year (y) | 532 (62.81%) |
| W1 Employment (y) | 824 (97.28%) |
| W1 Precarious Health (y) | 186 (21.96%) |
| W5 Self-rated Health | 3.53 (0.95) |
| W1 DAI Acceptance | 14.47 (5.60) |
| W1 DAI Externally Generated | 10.41 (3.80) |
| W1 DAI Finality | 10.12 (4.46) |
| W1 DAI Thoughts | 6.85 (3.16) |
| W1 Extraversion | 5.71 (1.94) |
| W1 Agreeableness | 6.82 (1.63) |
| W1 Conscientiousness | 7.65 (1.72) |
| W1 Neuroticism | 5.45 (2.10) |
| W1 Openness | 6.51 (1.72) |
| W1 Loneliness | 4.53 (1.80) |
| W1 Self-Esteem | 4.14 (1.54) |
| W1 Resilience | 20.20 (5.05) |
| W1 Intolerance Uncertainty | 34.27 (8.96) |
| W1 Trust in People (y) | 502 (62.05%) |
| W1 Neighbourhood Belongingness (y) | 527 (62.22%) |
| W1 Neighbourhood Connectedness (y) | 506 (59.74%) |
| W1 Neighbourhood Belonging Wider Community (y) | 428 (50.53%) |
| W3 Social Support Instrumental | 13.35 (5.72) |
| W3 Social Support Emotional | 13.59 (5.22) |
| W5 Social Engagement | 2.71 (1.31) |
| W1 Religious (y) | 534 (63.05%) |
| W1 Hygiene Behaviours | 64.24 (8.71) |
| W1 Social Distance | 65.09 (9.35) |
| W1 COVID-19 Anxiety | 67.79 (24.47) |
| W1 PTSD Symptoms | 3.57 (5.27) |
| W5 COVID-19 Exposure (y) | 692 (81.70%) |
| W5 Service Use (y) | 520 (61.40%) |

**Note:** W1 = Wave 1; W3 = Wave 3; W5 = Wave 5; y = yes; DAI = death anxiety inventory; PTSD = post traumatic stress disorder.

with *Resilient* outcomes. In the newly added COVID-block, *PTSD symptoms* (OR = 0.89, CI = 0.84–0.94, p < .001) were significantly and negatively associated with *resilient* outcomes whilst *service use* was positively associated with state *Resilient* outcomes at Wave 5 (OR = 2.34, CI = 1.48–3.74, p < .001). Fig 5 shows the plot of the odds ratio of the longitudinal model in the UK.

**Table 4. Results of the longitudinal regression model based on the UK data (N = 847).**

| Variable | Block 1 | | | + Block 2 | | | + Block 3 | | | + Block 4 | | | + Block 5 | | |
|---|---|---|---|---|---|---|---|---|---|---|---|---|---|---|---|
| | OR | 95% CI | p | OR | 95% CI | p | OR | 95% CI | p | OR | 95% CI | p | OR | 95% CI | p |
| W1 Gender binary | 0.70 | 0.49, 0.99 | **.046** | 0.73 | 0.47, 1.13 | .156 | 0.69 | 0.44, 1.09 | .114 | 0.70 | 0.44, 1.10 | .122 | 0.77 | 0.47, 1.26 | .301 |
| W1 Age | 1.03 | 1.02, 1.04 | **<.001** | 1.00 | 0.99, 1.02 | .725 | 1.00 | 0.98, 1.02 | .803 | 1.00 | 0.98, 1.02 | .957 | 0.99 | 0.97, 1.01 | .547 |
| W1 Education ≥ h.s. | 0.95 | 0.65, 1.37 | .776 | 1.03 | 0.65, 1.63 | .885 | 0.96 | 0.59, 1.56 | .872 | 0.97 | 0.59, 1.57 | .901 | 1.03 | 0.62, 1.70 | .920 |
| W1 Living alone | 0.77 | 0.49, 1.21 | .251 | 0.80 | 0.46, 1.40 | .422 | 1.13 | 0.59, 2.17 | .720 | 1.07 | 0.56, 2.06 | .840 | 0.95 | 0.49, 1.88 | .888 |
| W1 Children in household | 1.07 | 0.68, 1.70 | .776 | 1.20 | 0.70, 2.10 | .514 | 1.32 | 0.75, 2.38 | .344 | 1.34 | 0.76, 2.42 | .318 | 1.44 | 0.79, 2.67 | .238 |
| W1 House Property | 0.93 | 0.59, 1.43 | .732 | 0.83 | 0.48, 1.40 | .481 | 0.73 | 0.41, 1.28 | .281 | 0.74 | 0.41, 1.30 | .295 | 0.70 | 0.38, 1.27 | .247 |
| W1 Income | 1.54 | 1.04, 2.29 | **.032** | 0.86 | 0.53, 1.39 | .546 | 0.81 | 0.48, 1.36 | .433 | 0.78 | 0.46, 1.31 | .352 | 0.68 | 0.39, 1.18 | .173 |
| W1 Employment (y) | 2.75 | 1.12, 6.69 | **.025** | 1.24 | 0.40, 3.89 | .713 | 1.55 | 0.46, 5.25 | .478 | 1.58 | 0.47, 5.30 | .456 | 1.54 | 0.43, 5.54 | .504 |
| W1 Precarious Health (y) | 0.62 | 0.42, 0.94 | **.023** | 0.89 | 0.54, 1.47 | .641 | 0.77 | 0.46, 1.31 | .325 | 0.77 | 0.46, 1.32 | .340 | 0.93 | 0.54, 1.63 | .801 |
| | | | | + Block 2 | | | + Block 3 | | | + Block 4 | | | + Block 5 | | |
| | | | | OR | 95% CI | p | OR | 95% CI | p | OR | 95% CI | p | OR | 95% CI | p |
| **W5 Self health** | - | - | - | 1.87 | 1.47, 2.39 | **<.001** | 1.75 | 1.36, 2.25 | **<.001** | 1.77 | 1.38, 2.28 | **<.001** | 1.89 | 1.44, 2.50 | **<.001** |
| W1 DAI Acceptance | - | - | - | 0.99 | 0.92, 1.06 | .716 | 0.98 | 0.91, 1.05 | .532 | 0.97 | 0.91, 1.05 | .454 | 0.97 | 0.90, 1.05 | .406 |
| W1 DAI Externally Generated | - | - | - | 1.06 | 0.99, 1.15 | .112 | 1.06 | 0.98, 1.15 | .158 | 1.06 | 0.98, 1.15 | .129 | **1.09** | **1.00, 1.19** | **.047** |
| W1 DAI Finality | - | - | - | 0.96 | 0.89, 1.04 | .299 | 0.97 | 0.89, 1.05 | .425 | 0.97 | 0.90, 1.06 | .534 | 0.95 | 0.87, 1.04 | .272 |
| W1 DAI Thoughts | - | - | - | **0.91** | **0.83, 1.00** | **.043** | 0.92 | 0.84, 1.01 | .097 | 0.92 | 0.84, 1.01 | .086 | 0.99 | 0.89, 1.10 | .866 |
| W1 Extraversion | - | - | - | 0.98 | 0.86, 1.12 | .815 | 0.96 | 0.84, 1.11 | .605 | 0.98 | 0.85, 1.12 | .740 | 0.99 | 0.86, 1.15 | .891 |
| W1 Agreeableness | - | - | - | 0.99 | 0.86, 1.14 | .885 | 0.97 | 0.83, 1.13 | .713 | 0.98 | 0.84, 1.14 | .803 | 1.00 | 0.85, 1.17 | >.999 |
| W1 Conscientiousness | - | - | - | 1.02 | 0.89, 1.16 | .772 | 1.01 | 0.88, 1.16 | .862 | 1.01 | 0.88, 1.16 | .839 | 1.02 | 0.88, 1.18 | .768 |
| W1 Neuroticism | - | - | - | **0.85** | **0.73, 0.99** | **.037** | **0.82** | **0.70, 0.96** | **.016** | **0.82** | **0.70, 0.97** | **.018** | 0.85 | 0.72, 1.01 | .064 |
| **W1 Openness** | - | - | - | 1.12 | 0.98, 1.27 | .090 | 1.12 | 0.99, 1.28 | .080 | 1.12 | 0.99, 1.28 | .084 | **1.17** | **1.02, 1.34** | **.028** |
| **W1 Loneliness** | - | - | - | **0.76** | **0.67, 0.86** | **<.001** | **0.75** | **0.66, 0.85** | **<.001** | **0.75** | **0.66, 0.86** | **<.001** | **0.79** | **0.68, 0.90** | **<.001** |
| **W1 Self Esteem** | - | - | - | 1.19 | 1.00, 1.41 | .050 | 1.18 | 0.99, 1.42 | .072 | 1.18 | 0.99, 1.42 | .067 | **1.30** | **1.08, 1.58** | **.007** |
| W1 Resilience | - | - | - | 1.05 | 0.99, 1.12 | .132 | 1.05 | 0.99, 1.12 | .134 | 1.05 | 0.98, 1.12 | .147 | 1.04 | 0.97, 1.11 | .286 |
| W1 Intolerance Uncertainty | - | - | - | 0.97 | 0.94, 1.00 | .072 | 0.98 | 0.95, 1.01 | .248 | 0.98 | 0.95, 1.01 | .246 | 0.99 | 0.96, 1.02 | .550 |
| | | | | | | | + Block 3 | | | + Block 4 | | | + Block 5 | | |
| | | | | | | | OR | 95% CI | p | OR | 95% CI | p | OR | 95% CI | p |
| W1 Trust people (y) | - | - | - | - | - | - | 1.12 | 0.70, 1.80 | .623 | 1.10 | 0.68, 1.76 | .693 | 1.03 | 0.63, 1.70 | .893 |
| **W1 Neigh Belongingness (y)** | - | - | - | - | - | - | **1.67** | **1.03, 2.72** | **.038** | **1.69** | **1.04, 2.76** | **.035** | **1.96** | **1.18, 3.30** | **.010** |
| W1 Neigh Connectedness (y) | - | - | - | - | - | - | 1.14 | 0.69, 1.87 | .604 | 1.18 | 0.71, 1.93 | .525 | 1.21 | 0.72, 2.03 | .475 |
| W1 Neigh Belong Wider Com (y) | - | - | - | - | - | - | 0.89 | 0.53, 1.48 | .645 | 0.89 | 0.53, 1.50 | .660 | 0.93 | 0.53, 1.60 | .788 |
| W3 Soc Supp Instrumental | - | - | - | - | - | - | **1.08** | **1.01, 1.16** | **.036** | **1.08** | **1.00, 1.16** | **.044** | 1.07 | 0.99, 1.16 | .096 |

*(Continued)*

**Table 4.** (Continued)

| | | | | | | | | | | | | | | | |
|---|---|---|---|---|---|---|---|---|---|---|---|---|---|---|---|
| W3 Soc Supp Emotional | - | - | - | - | - | - | 0.95 | 0.87, 1.02 | .165 | 0.95 | 0.87, 1.02 | .173 | 0.97 | 0.89, 1.06 | .500 |
| W5 Social Engagement | - | - | - | - | - | - | 1.07 | 0.90, 1.27 | .454 | 1.07 | 0.90, 1.27 | .461 | 1.11 | 0.92, 1.33 | .260 |
| | | | | | | | | | | + Block 4 | | | + Block 5 | | |
| | | | | | | | | | | OR | 95% CI | p | OR | 95% CI | p |
| W1 Religion (y) | - | - | - | - | - | - | | | | 0.68 | 0.42, 1.09 | .110 | 0.77 | 0.46, 1.25 | .294 |
| | | | | | | | | | | | | | + Block 5 | | |
| | | | | | | | | | | | | | OR | 95% CI | p |
| W1 Hygiene | - | - | - | - | - | - | - | - | - | - | - | - | 0.99 | 0.95, 1.03 | .479 |
| W1 Social Distance | - | - | - | - | - | - | - | - | - | - | - | - | 1.01 | 0.97, 1.05 | .534 |
| W1 COVID19 Anxiety | - | - | - | - | - | - | - | - | - | - | - | - | 0.99 | 0.98, 1.00 | .240 |
| **W1 PTSD Symptoms** | - | - | - | - | - | - | - | - | - | - | - | - | **0.89** | **0.84, 0.94** | **< .001** |
| W5 COVID Exposure | - | - | - | - | - | - | - | - | - | - | - | - | 0.56 | 0.29, 1.04 | .074 |
| **W5 Service use** (y) | - | - | - | - | - | - | - | - | - | - | - | - | **2.34** | **1.48, 3.74** | **< .001** |

Note: OR = odds ratio; CI = 95% confidence interval; p = p value

*3.2.2.1 Resources predicting resilience or non-resilience.* Regarding the fourth research question, the resources that significantly predicted *Resilient* outcomes at W5 in the longitudinal model were, in order of strength: *service use* (access to health services was associated with *Resilient* outcomes) (OR = 2.34, CI = 1.47–3.37, p < .001), *belongingness to wider neighbourhood* (greater sense of belonging was associated with *Resilient* outcomes) (OR = 1.96, CI = 1.18–3.30; p = .010), *self-rated health* (more positive *self-rated health* was associated with *Resilient* outcomes) (OR = 1.89, CI = 1.44–2.50, p < .001), *self-esteem* (higher *self-esteem* was associated with *Resilient* outcomes) (OR = 1.30, CI = 1.08–1.58, p = .007), *Openness* (greater openness was associated with *Resilient* outcomes) (OR = 1.17, CI = 1.02–1.34, p = .028), *DAI Externally Generated* (higher levels of *Externally Generated DAI* was associated with *Resilient* outcomes) (OR = 1.09, CI = 1.00–1.19, p = .047).

Fig 6 shows a plot of the odds ratio of the resources (significant predictors) associated with *Resilient* outcomes in the longitudinal model. The predictors are ordered according to their strength in positively predicting resilience at W5, those in red are negatively associated with resilience–*PTSD symptoms* (OR = 0.89, CI = 0.84–0.94, p < .001) and *loneliness* (OR = 0.79, CI = 0.68–0.90, p < .001).

## 4. Discussion

This study aimed to further examine the effectiveness of the Ecological Model of Resilience (EMR) in explaining resilient outcomes during the COVID-19 pandemic in the UK. First, a cross-sectional model replicated the Italian study at Wave 3 in the UK. Second, going beyond the Italian cross-sectional study, a longitudinal regression model including additional EMR specific variables explored the predictors of resilience at Wave 5 using Wave 1, and with variables from Waves 3 and 5.

With respect to the first research question, at Wave 3 in the UK, 72% of participants were resilient, a finding in line with the Italian data, whilst at Wave 5, 80% were classified resilient. The Wave 3 findings were in line with the proportion of Italian participants who were resilient. Interestingly, the proportions of UK participants who were resilient increased between Wave 3 and Wave 5. In both cases most participants were just emerging from lockdowns. The increase

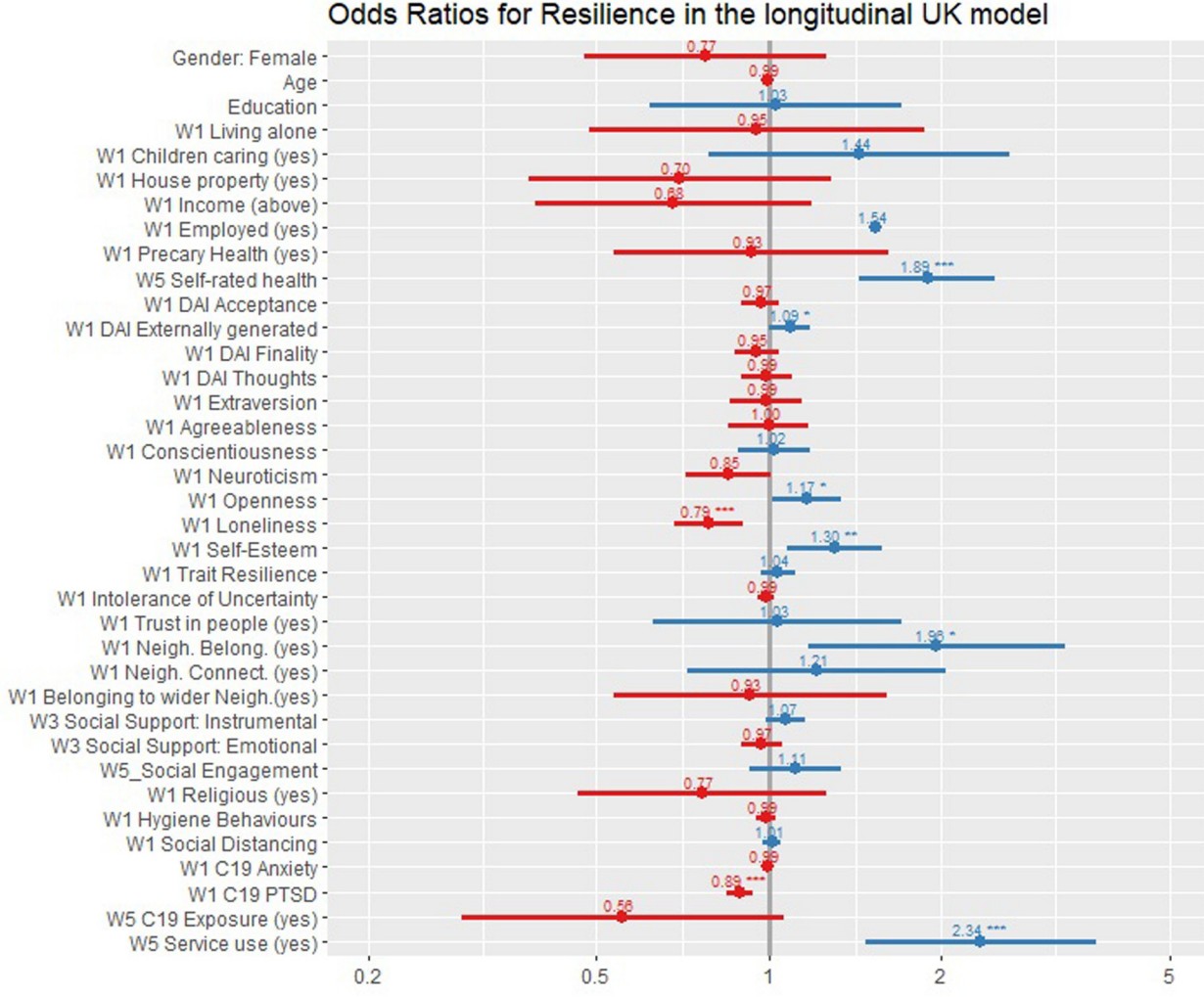

**Fig 5.**

in resilience suggests that there was some habituation to the effects of the pandemic. It is also possible that participants had developed successful coping strategies not measured in this study. It is also important to note that although the Wave 3 sample (N = 2019) was nationally representative this was not the case for the longitudinal study sample (N = 848) which included only those who had previously completed W1 and W3. As attrition naturally occurs in longitudinal studies, the longitudinal sample was not nationally representative [33, 51] and underrepresented younger people who have been more impacted by COVID-19 than older people [52, 53].

## The cross-sectional model in the UK

The results of the cross-sectional model in the UK (W3) successfully explained 67% of variance at W3. *Self-esteem* and *age* were the only significant predictors positively associated with *Resilient* outcomes (W3), whilst the other significant predictors were negatively associated with resilience and included *DAI Thoughts*, *Extraversion*, *PTSD symptoms*, *loneliness* and *Neuroticism*. The UK cross sectional study partially replicated the Italian results. Predictors in both studies were broadly in the same direction, that is either positively or negatively associated

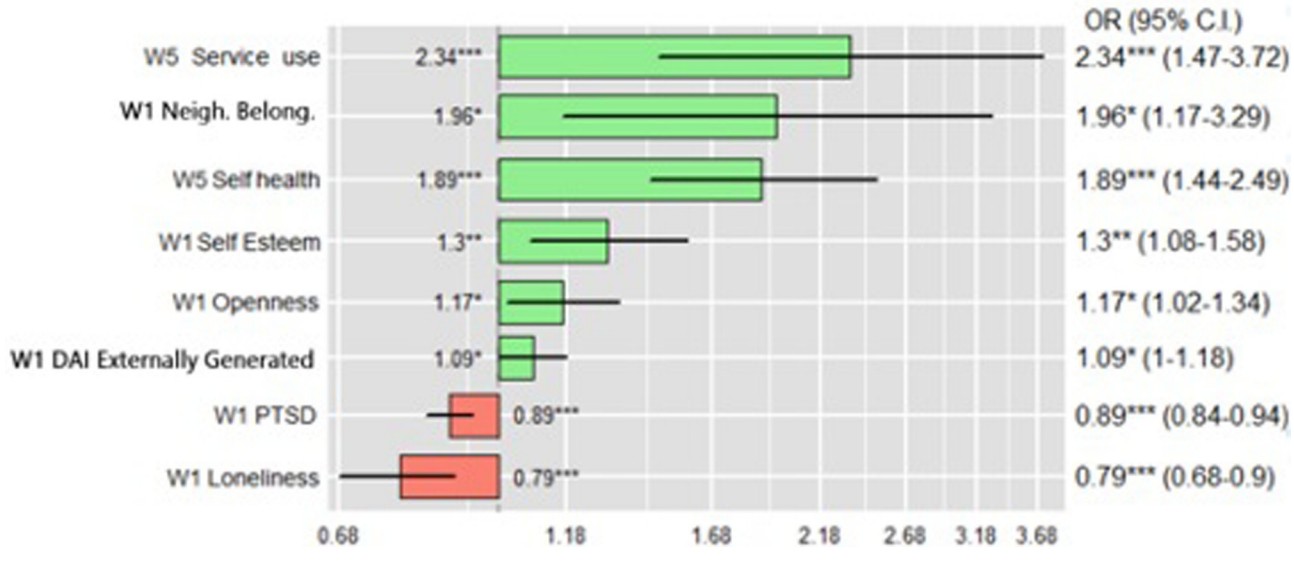

**Fig 6.**

with resilience. Non-resilient participants in both countries showed higher levels of *loneliness*, higher levels of COVID-related *PTSD symptoms* [54], higher *DAI Thoughts* [55] and higher *Neuroticism* [56] than Resilient participants–these represent cross-cultural risk factors, in line with existing literature.

However, there were variables which were significant in one study and not in the other. Notably, *self-esteem*, *age* (associated with *Resilience*) and *Extraversion* (associated with *Non-Resilience*) in the UK, and in Italy *caring for children*, *precarious health* and *Intolerance of Uncertainty* were associated with *Non-Resilient* outcomes, also *Conscientiousness* and *social distancing* were associated with *Resilient outcomes*. However, the *age* effect needs to be treated cautiously as the confidence intervals are near the non-effect boundary and is similar to the Italian estimate. *Intolerance of Uncertainty* was present only in the Italian study and not in the cross-sectional UK data at W3. There is some evidence that the UK has an *uncertainty approaching* culture whilst Italy has *uncertainty avoidant* one—for some considerations see [57, 58]. It is unclear why there are other different patterns regarding psychological factors for the two countries, although there may be cultural and contextual factors which may be influential but which we have not measured. This may also be the case for *caring for children* where there are likely to be both cultural and both general and COVID-19-specific contextual factors. In recent decades in Italy, family units have become progressively smaller, without relatives able to care for children [59, 60]. In addition, children in recent years are less likely to play unsupervised in the neighbourhood. These factors were compounded during the pandemic. In both countries, children could no longer play out, grandparents were isolating. In addition, children needed to be home schooled, and parents often juggled working from home and home schooling. In the UK children of key workers were able to attend school. In Italy parents often had to take unpaid leave to care for the children.

Comparing the UK and Italy, most predictors of resilience were consistent in the direction of their effect. However, none were statistically significant in both countries. This may suggest a concordant role for those predictors despite not reaching statistical significance. In the UK higher *age* was significant in positively predicting *Resilient* outcomes but on in Italy. In Italy, but not in the UK, *DAI Finality* and *Conscientiousness* were significantly positively associated

with *Resilient* outcomes. On the other hand, *caring for children* was significantly negatively associated with *Resilient* outcomes in Italy but not in the UK. There were only two significant predictors that were not directionally consistent between the two countries. *Extraversion* was significantly negatively associated with *Resilient* outcomes in the UK and *self-esteem* was positively associated with *Resilient* outcomes also in the UK but in Italy the results were non-significant. It is possible that the social restrictions imposed during the lockdown might have a greater impact on those with higher extraversion, than those with lower. However, it is not clear why that should be the case in the UK but not in Italy. It is noteworthy, that when comparing the UK and Italian cross-sectional EMRs, there were no predictors that were both statistically significant and in divergent directions, indicating a generally consistent pattern.

## The longitudinal model in the UK

Regarding the longitudinal model in the UK (RQ3), the results explained 47% of the variance of *Resilient* outcomes at W5 (compared to 67% of UK-W3). The predictors significantly associated with resilience in the UK longitudinal model were in the majority of resources with a positive association with *Resilient* outcomes. These protective factors included *service use*, *belongingness to neighbourhood*, *self-rated health*, *self-esteem*, *Openness*, and *Death Anxiety Externally Generated*. The two statistically significant predictors negatively associated with *Resilient outcomes* were *PTSD symptoms* and *loneliness*.

The positive association between *service use* and *Resilient* outcomes emphasises the importance of a well-functioning health system in times of health emergencies. Those that had access to, or did not need health services, were twice as likely to be *Resilient* than those who needed a service but could not access it. Thus, needing a service but not being able to access presents serious difficulties for people with mental/physical health issues. Our finding underlies the importance–at a societal level–of providing and guaranteeing essential health services during health-related emergencies, regardless of the pressure to restrict routine medical checks.

*Belongingness to neighbourhood* also promotes Resilience, providing people with a sense of connection to their communities which can be drawn on in times of need. Those with higher belongingness were also almost twice as likely to be *Resilient*. This is consistent with a wide range of evidence that identification with groups is facilitates mental health and protects against threats to wellbeing to the extent to which this process has been described as a 'social cure' [61]. Individuals reporting higher *belongingness to neighbourhood* may also perceive higher social support (both instrumental and emotional), an important resource for psychological health [62].

Among the psychological variables, the strongest predictor of *Resilience* was *self-rated health*. *Self-rated health* which, in previous research, has been found to be an important predictor of mortality [63]. Indeed, individuals with overall better subjective health draw on stronger perceived foundations (e.g., strength, wellness, physical functioning) in facing difficulties (both practical and psychological). Individuals with higher *self-rated health* are also less worried about physical issues that are known to negatively affect psychological health and quality of life, especially during health-related emergencies [64, 65]. In addition, this result is in line with research which finds that *self-rated health* reflects individual psychological health [26].

*Self-esteem* also promoted *Resilient* outcomes. Having a positive idea, representation, and evaluation of oneself may be an advantage for several reasons [66], since it allows the experience of more confidence, acceptance, and unconditional appreciation toward self. Positive self-evaluation is often associated with a positive view of the world, life and the future, allowing being more optimistic and feeling in control of one's own life. As a result, people with higher *self-esteem* are less prone to a negative cognitive style–(i.e., negative events interpretations)–

and less prone to experience negative affectivity [67]. Research on Terror Management Theory (TMT) [68], which focuses on how individuals cope with knowledge of mortality, supports this finding. According to TMT, in threatening contexts such as the COVID-19 pandemic [69], *self-esteem* acts as a buffer, shielding individuals from anxiety and depression [70].

Among the personality traits, *Openness to Experience* was a protective factor promoting *Resilience*, Literature shows the positive associations among *Openness to Experience*, positive emotions, quality of life, and well-being [71]. Conversely, *Openness to Experience* is negatively related to negative affect [72]. This variable may be associated with psychological flexibility which is associated with good mental health [64, 73, 74]. Further, a lack of psychological flexibility is a well-known risk-factor for psychological difficulties, intolerance of uncertainty and the need for closure [75–79].

Lastly, the *Externally Generated Death Anxiety*–consisting in being upset when facing external stimuli related to death (e.g., coffins, cemeteries)–showed a significant positive association with *Resilient* outcomes. Fearing death-related *stimuli* can be considered as a cognitive avoidance process [75]. According to the cognitive avoidance theory [80], the visual exposure (real or imaginary) to triggering stimuli generates a strong physiological activation, favouring emotional processing, thus ultimately promoting lower levels of depression and anxiety over time (i.e., through the pandemic). Conversely, cognitive avoidance/suppression uses verbalization as a strategy for abstraction, disengagement, and emotion control, to lower the physiological activation to aversive material. In a nutshell, abstract verbal thoughts are less emotionally activating than visual images (e.g., corpses, blood) so that, in the short-term, abstract thought prevents emotional elaboration but in the mid-long term can lead to emotional disturbance [81]. Similarly, in a first phase of the pandemic, high levels of *externally generated anxiety* when visualizing of death-related stimuli reflected the heightened psycho-physiological activation that promoted emotional processing [82] and favoured resilience in the long term.

Regarding the risk factors hindering *Resilient outcomes*, *loneliness* was the strongest one followed by *PTSD symptoms*. The COVID-19 pandemic exacerbated feelings of *loneliness* feelings through the forced isolation and physical distancing that weakened social connections [83, 84]. Loneliness is a well-known risk factor for mental ill-health, since it can trigger distressing thoughts focusing on comparisons between the actual and the desired socio-relational situation, leading to strong symptoms of anxiety that–in turn–can trigger depressive symptoms [85–87]. Moreover, this finding is also congruent and complementary to the above-discussed association of *belongingness to neighbourhood* with Resilience.

Lastly, *PTSD symptoms* related to COVID-19 were a risk-factor for *Non-Resilient* outcomes, suggesting how the COVID-19 has been perceived as a traumatic event concretely threatening the brief and long-term health of self and loved ones. *PTSD symptoms* are characterised by three areas consisting in frequent negative emotions, heightened arousal and re-experiencing of the traumatic events—all together these characteristics lead to higher anxious and depressive symptoms resulting in *Non-Resilient* outcomes.

It was notable that *Openness to Experience*, *loneliness*, *self-esteem* and *PTSD symptoms*–all measured at W1 predicted *Resilient* or *Non-Resilient* outcomes at W5, suggesting that interventions for prevention and treatment of psychological issues should consider these indicators. Interestingly, *Openness to Experience* was not a significant predictor at W3 but it was from W1 to W5, possibly suggesting that the abilities to cope with the situation and elaborate it may require a certain time to be effective [88, 89].

Among the significant predictors from W1 to W5, the majority were dispositional-like traits—as *self-esteem* and *Openness to Experience*. *Openness to Experience* is clear is a personality trait and therefore likely stable across time but this is less certain for *self-esteem*. However, the available research literature suggests that self-evaluation tends to be stable over time to the

extent to which it is grounded in early personal relationships and then reflected stable behavioural patterns [90]. It is not surprising that trait-like variables are associated with state resilience over time, since their stable nature allows scores to be more consistent over time when compared to state-like characteristics that are more prone to change. Although trait-like factors have traditionally been seen as more difficult to modify, recent third Wave psychotherapy approaches are having some success [91, 92].

It is interesting to consider how the positive associations with Resilient Outcomes might be utilised to mitigate against the Non-Resilient Outcomes. For example, interventions which promote *self-esteem* and increase *Belongingness to Neighbourhood* might reduce *loneliness* [93]. Similarly, access to services is an important resource for people with *PTSD symptoms*, and with COVID-related *PTSD symptoms*. Policy makers should consider ways of keeping services open during times of pandemic. Increasing *Belongingness to Neighbourhood* might also enable people with PTSD to engage with support from local communities.

Although the focus of this paper is on Resilient outcomes, it is important to note that 20% of the sample were not resilient. These participants had reached either, or indeed both, the threshold for clinical caseness for either anxiety or depression. Much of the focus of research during the pandemic has focused on these groups (e.g. 3). Shevlin et al.'s [94] one-year follow-up analysis using the C19PRC dataset found that approximately 5% of the sample had deteriorating mental health associated with the pandemic. Future research which focuses on facilitating resilience amongst this population in the face of significant societal challenge would be particularly useful.

This research is not without limitations. Even though the W1 and W3 samples were balanced and stratified to reflect the population characteristics, the internet-administered survey may have reached a population that is friendly with technology use. Further, the sample is less representative in the longitudinal analyses due to attrition over time. For a discussion of the limitations of the C19PRC study see for Wave 1 and 2 [6], Wave 3 [31], and Wave 5 [33]. When interpreting the results, we should be aware that there may be other variables which we have not measured which may impact on resilience outcomes. There may also be high-order interactions which should be explored in future research.

Despite these limitations, this research has considerable strengths. As part of an international C19PRC project, the structure of the panel study allows several variables to be measured across different areas (psychological, community, societal, COVID-related), at different points of time, and across different countries. The replication of the Italian resilience study [26] allowed cross-cultural comparison of the ERM. In addition, the flexible nature of the data collection in the UK enabled us to refine the data collection to include variables relevant to the model such as service use, social support and social engagement [21, 95]. This research also provides validation of results in two distinct cohorts.

Future research will test the validity of the EMR in other countries to extend the impact of this research. Further studies may also enable us to understand further cross-cultural differences in resilience. Lastly, studying the trajectories of resilience over time is a complex topic requiring further attention and comprehension which will lead to more effective interventions.

In conclusion, the EMR provided a useful framework for explaining and understanding resilience and the role of vulnerability factors and resources in the COVID-19 pandemic in the UK. Further, the model identifies individual, community and societal resources which are open to modification to enhance resilience.

## Author Contributions

**Conceptualization:** Kate M. Bennett, Anna Panzeri, Sarah Butter, Todd K. Hartman, Jamie Murphy, Mark Shevlin, Jilly Gibson-Miller, Anton P. Martinez, Thomas V. A. Stocks.

**Data curation:** Sarah Butter.

**Formal analysis:** Anna Panzeri.

**Investigation:** Kate M. Bennett, Anna Panzeri, Elfriede Derrer-Merk, Liam Mason, Mark Shevlin, Jilly Gibson-Miller, Liat Levita, Anton P. Martinez, Ryan McKay, Gioa Bottesi, Richard P. Bentall.

**Methodology:** Kate M. Bennett, Anna Panzeri, Giulo Vidotto, Marco Bertamini.

**Project administration:** Kate M. Bennett.

**Resources:** Kate M. Bennett.

**Supervision:** Todd K. Hartman, Liam Mason, Orla McBride, Jamie Murphy, Giulo Vidotto, Richard P. Bentall, Marco Bertamini.

**Visualization:** Kate M. Bennett, Anna Panzeri.

**Writing – original draft:** Kate M. Bennett, Anna Panzeri, Marco Bertamini.

**Writing – review & editing:** Kate M. Bennett, Anna Panzeri, Elfriede Derrer-Merk, Sarah Butter, Todd K. Hartman, Liam Mason, Orla McBride, Jamie Murphy, Mark Shevlin, Jilly Gibson-Miller, Liat Levita, Anton P. Martinez, Ryan McKay, Alex Lloyd, Thomas V. A. Stocks, Gioa Bottesi, Giulo Vidotto, Richard P. Bentall, Marco Bertamini.

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
