## [Decision Letter · Decision Letter 0]

26 Oct 2022

PONE-D-22-23187Predicting Resilience during the COVID-19 Pandemic in the United Kingdom: Cross- sectional and Longitudinal ResultsPLOS ONE

Dear Dr. Bennett,

Thank you for submitting your manuscript to PLOS ONE. After careful consideration, we feel that it has merit but does not fully meet PLOS ONE’s publication criteria as it currently stands. Therefore, we invite you to submit a revised version of the manuscript that addresses the points raised during the review process.

We look forward to receiving your revised manuscript.

Kind regards,

Professor Lambros Lazuras

Academic Editor

PLOS ONE

Journal Requirements:

6. Please ensure that you refer to Figure 2 in your text as, if accepted, production will need this reference to link the reader to the figure.

Additional Editor Comments:

Three independent reviewers thoroughly reviewed your manuscript. One suggested acceptance and the second minor revisions, whereas the third reviewer suggested major revisions. Following my own reading of the manuscript I concur with the first 2 reviewers and recommend minor revisions.

Reviewers' comments:

Reviewer's Responses to Questions

**Comments to the Author**

1. Is the manuscript technically sound, and do the data support the conclusions?

Reviewer #1: Yes

Reviewer #2: Yes

Reviewer #3: Partly

2. Has the statistical analysis been performed appropriately and rigorously? 

Reviewer #1: Yes

Reviewer #2: Yes

Reviewer #3: Yes

3. Have the authors made all data underlying the findings in their manuscript fully available?

Reviewer #1: Yes

Reviewer #2: Yes

Reviewer #3: No

4. Is the manuscript presented in an intelligible fashion and written in standard English?

Reviewer #1: Yes

Reviewer #2: Yes

Reviewer #3: Yes

5. Review Comments to the Author

Reviewer #1: The study investigates the effects of psychological and social factors on resilience in a longitudinal study across the UK and Italy. The study data are part of a larger study of an international research consortium (COVID-19 Psychological Research Consortium (C19PRC). The study is written in an excellent, concrete and clear manner. The design and analysis of the data re robust and well-presented. The figures, tables and graphs are informative and address the RQs. The discussion is clear and informative and the conclusions are appropriate. The manuscript could be improved in two main areas:

1. A comprehensive description of "resilience" is missing form the introduction. Although the stance taken in this paper is clearly articulated (low anxiety/depression), there is a need for an overview of resilience as a psychological construct, and how this is depicted in crises (please include relevant examples from other studies - perhaps of natural disasters etc)

2. A more comprehensive discussion of what the findings mean in the context of psychological well being. Approx. 80% of the participants showed resilience in the UK in Wave 5, but what about the 20%, what does it mean in terms societal challenges in the future.

3. A more comprehensive discussion on the negative correlates of resilience (PTSD and loneliness) and the how these barriers to resilience can be overcome - using data from the positive correlates of resilience.

Reviewer #2: I would thank you very much for the opportunity of revising the manuscript.

Abstract

Minor revisions:

Line 58: please, be consistent in reporting "Covid-19" using the uppercase.

Line 60: please, delete the acronym "W3", there is no reason to report it.

Line 61: in my opinion, it is not clear the reason why the authors reported the N referring only to a sample.

Line 62: please, report the full name of the C19PRC or, if you prefer, report this acronym previous, specifically where you mention it for the first time.

Line 67: please, do not report the acronym you never mentioned previously (especially, in the abstract section).

Introduction

The introduction is well written and clear in terms of context, theoretical framework, aims, and scopes of the present study. Moreover, the research questions are well explained, and it is clear the contribution of the present study.

Minor revisions

Line 100: please, report the acronym of Ecological Model of Resilience when you mention the model for the first time if you use EMR along the manuscript (please, see line 114).

Figure 1: could you insert a figure with higher quality? The figure is clear in terms of theoretical reference, but the quality is not excellent.

Lines 116-117: please, could you report the acronym of PHQ and GAD specifying what they refer to? Those who read the manuscript for the first time do not immediately comprehend the meaning of these two acronyms. Moreover, please, do not use parentheses into parentheses, you should separate the text in the parentheses with semicolons.

Line 121: please, add a space between "(3)" and "again".

Line 134: please, add a come after "In the current study".

Line 139: I think you should add a point after "(July 2020)".

Materials and Methods

The methodological section is also well written. The authors well described all the measures, instruments, variables (i.e., both dependent and independent variables), and statistical analyses. Nevertheless, I suggest minor revisions, as follows:

Minor revisions

Line 152: I think you should report the word "wave" using "Wave", as you report in the introduction section. Please, report the present revision along all the manuscript (e.g., line 167).

Line 161: please, consider the same revision regarding the use of the parentheses in the text.

Line 166: please, add the equal symbol (=) between "mean age" and the value you reported.

Line 167: please, report what the value 51.46 refers to. It is the mean age, I guess.

Lines 166 – 169: it is not completely clear, in my opinion, if the final number of respondents in both samples (i.e., cross-sectional, and longitudinal studies)

Lines 177, 178, 180, 182, 186, 206, 2016, 235, 350, 510, 511, 564, etc.: please, consider the same revision regarding the use of the parentheses in the text.

Line 187, 192, and 194: please, be consistent in reporting the zero before the decimal point.

Line 215 - 238: please, could you report an item example for each measure?

Line 227: please, could you add the value of the good reliability and validity or the reference where readers can find this information?

Line 232: please, report the α-value for the SISES.

Lines 236 - 237: please, be consistent with previous measures' descriptions in reporting the response scale range.

Lines 255 - 256: please, be consistent with previous measures' descriptions in reporting the response scale range.

Lines 275 - 276: please, be consistent with previous measures' descriptions in reporting the response scale range.

Line 277: please, moderate the use of the parentheses and verify if you closed the parentheses.

Lines 282 - 283: please, be consistent with previous measures' descriptions in reporting the response scale range.

Line 287 and line 290: please, be consistent with previous measures' descriptions in reporting the response scale range.

Line 295: please, be consistent with previous measures' descriptions in reporting the response scale range.

Line 299: please, add a comma after "e.g.".

Line 300: please, be consistent with previous measures' descriptions in reporting the response scale range.

Line 306: please, be consistent with previous measures' descriptions in reporting the response scale range.

Line 319: please, delete the colon.

Line 335: please, use the Journal's style for reporting citations.

More in general, for all the α-values, please, be consistent in declaring the decimal number in terms of the numbers after the point.

Results

The results are well explained and described. I suggest, the following few modifications.

More in general, please, report the p-values using italics and be consistent in reporting the zero before the decimal numbers. Finally, be consistent in mentioning the name of the variables in terms of uppercase and lowercase.

Minor revisions

Line 349: please, consider deleting one of the two percentages regarding the gender- balance. It is extremely redundant.

Table 1: it is necessary to improve Table 1. Please, it could be necessary to align thousands and hundreds under the same column, use the same font, and add the meaning of all acronyms or abbreviations. I would also suggest not using all this white space.

Line 359: please, do not use italics for the words: "state", "Resilient" and "Non-Resilient", unless there is a reason.

Line 361: I suggest using the acronym ERM.

Table 2: please, add the "notes" to explain the acronyms (e.g., OR, CI, W, y, h.s., etc.) and, in my opinion, it is necessary to report the meaning of the colors (i.e., red and green).

Figure 2 and Figure 3: it is necessary to improve the quality of the imagines.

Figure 3: there are two Figures with the same name (i.e., Figure 3). Please, define better the Figure's names in order to distinguish them.

Table 3: the column with values describes only the number and the percentage for each variable, although the title of this column reports also the Mean and the SD. Please, modify the title or add this information. Moreover, please, add the "notes" to explain the acronyms (e.g., W, DAI, y, etc.).

Line 449: you report 846 participants. Is there an attrition rate? In previous paragraphs, you referred that the number of participants is 847.

Line 480: for me, it is not clear what "+0.003" represent.

Discussion

The discussion is well written, the aims of the study were well summarized, and the results were interestingly explained and discussed. Nevertheless, I highly recommend trying to reduce and streamline the text in order to obtain a text clearer and more pleasant. In this way, I strongly think that the reading would obtain a better flow.

Minor revision:

Line 535: please, add a comma after "UK".

Lines 547, and 589: in my opinion, it is not necessary to use the bold style in the text.

Lines 569 -570: please, revise the citation.

Line 608: please, add a space after the point.

Reviewer #3: The paper is an examining of factors that predict ‘resilience’ from a medium-sized convenience sample. The positives are the large range of psychometric measures, the careful modelling of groupings of variables and the validation of results in two distinct cohorts. The main negative is the extent that ‘resilience’ is well defined and we do not know how biased the sample is.

Before the ecological model of resilience is posited, it would be good to know what is meant by ‘resilience’

Reading on, it looks like resilience is just ‘good mental health’. In which case, I am not sure how the analysis differs from just looking at risk factors for mental health. Could you please clarify?

The term ‘clinical caseness’ is problematic because it implies a diagnosis, however a threshold from a questionnaire is not used to diagnose, by itself, without a clinical review. I suggest using another term.

It is not clear how the sample were recruited, apart from ‘online’. More detail is needed. Also, how representative were they, apart from the variables used to define the sampling strata?

I am not sure what is meant by the term “univariate multiple regression” – is this an adjusted model or not?

I am not clear why extraversion would be negatively associated with resilience, so defined.

6. PLOS authors have the option to publish the peer review history of their article (what does this mean?). If published, this will include your full peer review and any attached files.

Reviewer #1: No

Reviewer #2: **Yes: **Federica Galli

Reviewer #3: **Yes: **Matthias Pierce

---

## [Author Response · Author response to Decision Letter 0]

7 Dec 2022

Review comment and Response

Reviewer #1: 

The study is written in an excellent, concrete and clear manner. The design and analysis of the data re robust and well-presented. The figures, tables and graphs are informative and address the RQs. The discussion is clear and informative and the conclusions are appropriate.

The manuscript could be improved in two main areas:

1. A comprehensive description of "resilience" is missing form the introduction. Although the stance taken in this paper is clearly articulated (low anxiety/depression), there is a need for an overview of resilience as a psychological construct, and how this is depicted in crises (please include relevant examples from other studies - perhaps of natural disasters etc)

Have added

Resilience has been conceptualised in a number of ways, including as a trait (Rossi et al., 2007), as stability in wellbeing despite challenge (Galatzer-Levy & Bonanno, 2012), or as bouncing back from adverse life events (Bennett, 2010). We utilise Windle’s (2011) large scale concept analysis, and employ her definition of resilience: “Resilience is the process of negotiating, managing and adapting to significant sources of stress or trauma. Assets and resources within the individual, their life and environment facilitate this capacity for adaptation and bouncing back in the face of adversity. Across the life course, the experience of resilience will vary” (p. 163). Resilience differs from notions of good mental health in its central requirement for challenge or trauma, in the case of this study, living through the COVID-19 pandemic. Other contexts relevant to resilience include natural disasters (Heppenstall et al., 2013), bereavement (Boerner et al., 2005), dementia (Gaugler et al., 2007), and war (Kimhi et al., 2021). Cosco et al., (2017) reviews different measurement approaches, and in this study we operationalise resilience as the absence of both depressive and anxiety symptoms (below the criteria for clinical caseness) in the face of significant challenge, namely the COVID-19 pandemic.

2. A more comprehensive discussion of what the findings mean in the context of psychological wellbeing. Approx. 80% of the participants showed resilience in the UK in Wave 5, but what about the 20%, what does it mean in terms societal challenges in the future.

Added:

Although the focus of this paper is on Resilient outcomes, it is important to note that 20% of the sample were not resilient. These participants had reached either, or indeed both, the threshold for clinical caseness for either anxiety or depression. Much of the focus of research during the pandemic has focused on these groups (refs). Shevlin et al.’s (2021) one-year follow-up analysis using the C19PRC dataset found that approximately 5% of the sample had deteriorating mental health associated with the pandemic. Future research which focuses on facilitating resilience amongst this population in the face of significant societal challenge would be particularly useful. 

3. A more comprehensive discussion on the negative correlates of resilience (PTSD and loneliness) and the how these barriers to resilience can be overcome - using data from the positive correlates of resilience.

Added

It is interesting to consider how the positive associations with Resilient Outcomes might be utilised to mitigate against the Non-Resilient Outcomes. For example, interventions which promote self-esteem and increase Belongingness to Neighbourhood might reduce loneliness. Similarly, access to services is an important resource for people with PTSD symptoms, and with COVID-related PTSD symptoms. Policy makers should consider ways of keeping services open during times of pandemic. Increasing Belongingness to Neighbourhood might also enable people with PTSD to engage with support from local communities. 

Reviewer #2: 

I would thank you very much for the opportunity of revising the manuscript.

Abstract

Minor revisions:

Line 58: please, be consistent in reporting "Covid-19" using the uppercase.

Have addressed

Line 60: please, delete the acronym "W3", there is no reason to report it.

Have addressed

Line 61: in my opinion, it is not clear the reason why the authors reported the N referring only to a sample.

Have addressed

Line 62: please, report the full name of the C19PRC or, if you prefer, report this acronym previous, specifically where you mention it for the first time.

Answer: where spelled for the first time

Line 67: please, do not report the acronym you never mentioned previously (especially, in the abstract section).

Answer: the acronym was explained.

Introduction

The introduction is well written and clear in terms of context, theoretical framework, aims, and scopes of the present study. Moreover, the research questions are well explained, and it is clear the contribution of the present study.

Thank you.

Minor revisions

Line 100: please, report the acronym of Ecological Model of Resilience when you mention the model for the first time if you use EMR along the manuscript (please, see line 114).

Answer: we made consistent the use of the ERM acronym.

Figure 1: could you insert a figure with higher quality? The figure is clear in terms of theoretical reference, but the quality is not excellent.

The figure is prepared as a tif. When inserted into the document like this the quality is less good but will be in final document.

Lines 116-117: please, could you report the acronym of PHQ and GAD specifying what they refer to? Those who read the manuscript for the first time do not immediately comprehend the meaning of these two acronyms. Moreover, please, do not use parentheses into parentheses, you should separate the text in the parentheses with semicolons.

Answer: thank you for pointing out this, we spelled these acronyms and removed the nested parentheses.

Line 121: please, add a space between "(3)" and "again".

Answer: thanks, we did. 

Line 134: please, add a come after "In the current study".

Answer: thanks, we added the comma.

Line 139: I think you should add a point after "(July 2020)".

Answer: added, thanks.

Materials and Methods

The methodological section is also well written. The authors well described all the measures, instruments, variables (i.e., both dependent and independent variables), and statistical analyses. 

Thank you

Nevertheless, I suggest minor revisions, as follows:

Minor revisions

Line 152: I think you should report the word "wave" using "Wave", as you report in the introduction section. Please, report the present revision along all the manuscript (e.g., line 167).

Answer: thanks, we corrected it and made it consistent throughout the manuscript.

Line 161: please, consider the same revision regarding the use of the parentheses in the text.

Answer: thanks, this is due to the referencing formatting. We fixed it.

Line 166: please, add the equal symbol (=) between "mean age" and the value you reported.

Answer: thanks, we added it.

Line 167: please, report what the value 51.46 refers to. It is the mean age, I guess.

Answer, yes, it was. we added it.

Lines 166 – 169: it is not completely clear, in my opinion, if the final number of respondents in both samples (i.e., cross-sectional, and longitudinal studies)

Answer: the participants of W3 are 2019. Due to attrition rates, the participants who jointly completed W1, W3, and W5 are fewer

Lines 177, 178, 180, 182, 186, 206, 2016, 235, 350, 510, 511, 564, etc.: please, consider the same revision regarding the use of the parentheses in the text.

Answer: thanks, this is due to the referencing formatting. We fixed it.

Line 187, 192, and 194: please, be consistent in reporting the zero before the decimal point.

Answer: thanks, we removed the zero before the decimal point if the reported value was not supposed to overpass 1 (eg. probabilities, p values, Cronbach alpha, etc).

Line 215 - 238: please, could you report an item example for each measure?

Answer: yes, we added them.

Line 227: please, could you add the value of the good reliability and validity or the reference where readers can find this information?

Answer: yes, we did it.

Line 232: please, report the α-value for the SISES.

Answer: the SISES is a single item measure thus its Cronbach alpha value cannot be calculated.

Lines 236 - 237: please, be consistent with previous measures' descriptions in reporting the response scale range.

Lines 255 - 256: please, be consistent with previous measures' descriptions in reporting the response scale range.

Lines 275 - 276: please, be consistent with previous measures' descriptions in reporting the response scale range.

Lines 282 - 283: please, be consistent with previous measures' descriptions in reporting the response scale range.

Line 287 and line 290: please, be consistent with previous measures' descriptions in reporting the response scale range.

Line 295: please, be consistent with previous measures' descriptions in reporting the response scale range.

Line 300: please, be consistent with previous measures' descriptions in reporting the response scale range.

Line 306: please, be consistent with previous measures' descriptions in reporting the response scale range.

Answer: thanks, about the comments about scales response format, we prefer to leave as it is in order not to be repetitive and still convey the same information.

Line 277: please, moderate the use of the parentheses and verify if you closed the parentheses.

Answer: thanks, this is due to the referencing formatting. We fixed it.

Line 299: please, add a comma after "e.g.".

Answer: thanks, we added the comma.

Line 319: please, delete the colon.

Answer: we removed it, thanks.

Line 335: please, use the Journal's style for reporting citations.

Answer: thanks, we corrected it.

More in general, for all the α-values, please, be consistent in declaring the decimal number in terms of the numbers after the point.

Answer: thanks, for alpha we decided to stick to two numbers after the decimal point.

Results

The results are well explained and described. I suggest, the following few modifications.

More in general, please, report the p-values using italics and be consistent in reporting the zero before the decimal numbers. Finally, be consistent in mentioning the name of the variables in terms of uppercase and lowercase.

Answer: we did not report the zero in p values according to APA-7 guidelines because a probability cannot reach nor exceed 1.

Have attended to second point

Minor revisions

Line 349: please, consider deleting one of the two percentages regarding the gender- balance. It is extremely redundant.

Answer: thanks, we removed one.

Table 1: it is necessary to improve Table 1. Please, it could be necessary to align thousands and hundreds under the same column, use the same font, and add the meaning of all acronyms or abbreviations. I would also suggest not using all this white space.

Answer: thanks, we aligned the thousands and hundreds, used the same font, and spelled the acronyms. About the blank space, we know that the table will be reformatted in the proofing process. 

Line 359: please, do not use italics for the words: "state", "Resilient" and "Non-Resilient", unless there is a reason.

Answer: We have chosen to use Italics because we are referring to variables defined in this paper to ensure clarity, where sometimes in the literature there is a lack of precision

Line 361: I suggest using the acronym ERM.

Answer: thanks, we modified it.

Table 2: please, add the "notes" to explain the acronyms (e.g., OR, CI, W, y, h.s., etc.) and, in my opinion, it is necessary to report the meaning of the colors (i.e., red and green).

Answer: thanks. we explained all the acronyms and the meaning of the colours.

Figure 2 and Figure 3: it is necessary to improve the quality of the imagines.

Answer: As previously said, the figures when inserted into the document like this the quality is less good but will be in final document. We upload them separately in high resolution.

Figure 3: there are two Figures with the same name (i.e., Figure 3). Please, define better the Figure's names in order to distinguish them.

Answer: thank you for noticing this. we have renumbered it and the following figures.

Table 3: the column with values describes only the number and the percentage for each variable, although the title of this column reports also the Mean and the SD. Please, modify the title or add this information. Moreover, please, add the "notes" to explain the acronyms (e.g., W, DAI, y, etc.).

Answer: thanks, the column reports frequency (percentage%) for categorical variables or mean (standard deviation) for continuous variables (the questionnaire outcomes).

We added all the spelled acronyms in the note below the table. 

Line 449: you report 846 participants. Is there an attrition rate? In previous paragraphs, you referred that the number of participants is 847.

Answer: No, it was a typo. thanks for noticing it, we corrected it.

Line 480: for me, it is not clear what "+0.003" represent.

Answer: it is the delta in the pseudo R square comparing the current model vs the previous one. We added this information - thanks.

Discussion

The discussion is well written, the aims of the study were well summarized, and the results were interestingly explained and discussed. 

Answer: Thank you.

Nevertheless, I highly recommend trying to reduce and streamline the text in order to obtain a text clearer and more pleasant. In this way, I strongly think that the reading would obtain a better flow.

Answer: Thanks, we acknowledge that the manuscript is long but it conveys important information which would be lost, and defer to R1’s comments about the clarity of the paper.

Minor revision:

Line 535: please, add a comma after "UK".

Answer: thanks, we added it.

Lines 547, and 589: in my opinion, it is not necessary to use the bold style in the text.

Answer: thanks, we modified both.

Lines 569 -570: please, revise the citation.

Answer: thanks, we did it.

Line 608: please, add a space after the point.

Answer: thanks, we did it.

Reviewer #3:

 The paper is an examining of factors that predict ‘resilience’ from a medium-sized convenience sample. The positives are the large range of psychometric measures, the careful modelling of groupings of variables and the validation of results in two distinct cohorts. The main negative is the extent that ‘resilience’ is well defined and we do not know how biased the sample is.

Before the ecological model of resilience is posited, it would be good to know what is meant by ‘resilience’

Reading on, it looks like resilience is just ‘good mental health’. In which case, I am not sure how the analysis differs from just looking at risk factors for mental health. Could you please clarify?

Please see additional paragraph in response to R1

The term ‘clinical caseness’ is problematic because it implies a diagnosis, however a threshold from a questionnaire is not used to diagnose, by itself, without a clinical review. I suggest using another term.

Have added:

Note that whilst we use the clinical cut-offs for both PHQ- 9 and GAD-7, we are not using them for the purpose of clinical diagnosis.

It is not clear how the sample were recruited, apart from ‘online’. More detail is needed. Also, how representative were they, apart from the variables used to define the sampling strata?

Have added

Participants were recruited online, via Qualtrics, with stratified quota sampling to achieve representative sample in terms of age and sex (using 2016 population estimates form Eurostat (2020) and household income matched to Office of National Statistics data (for more details see (McBride et al., 2020). Qualtrics alerted participants to the study, via Qualtrics (see McBride et al., 2020, 2021a, b, 2022 for details of the C19PRC methodology).

I am not sure what is meant by the term “univariate multiple regression” – is this an adjusted model or not?

Answer: thanks for the comment, we further specify this point. It is a regression model with only one dichotomous dependent variable (resilient = 1 or not resilient = 0). The predictor variables are progressively entered in blocks.

The model is adjusted for confounders (since the first block, eg., age, gender) in the sense that each variable effect is estimated once the other variables’ effects are considered. Indeed, this regression uses the type III sum of squares so that each variables explains a part of variance, once the variance explained from the others is removed. However, if you mean adjusting by considering the interaction effects, we didn’t consider them given the high number of predictors and possible interactions.

We briefly added this information to the manuscript as follows: 

“Regressions allowed to estimate the effect of each predictor variable on the dependent variable once the effect of the other variables has been considered; in this model the interactions among variables were not considered.” 

I am not clear why extraversion would be negatively associated with resilience, so defined.

Have added

It is possible that the social restrictions imposed during the lockdown might have a greater impact on those with higher extraversion, than those with lower. However, it is not clear why that should be the case in the UK but not in Italy.

---

## [Decision Letter · Decision Letter 1]

6 Mar 2023

Predicting Resilience during the COVID-19 Pandemic in the United Kingdom: Cross- sectional and Longitudinal Results

PONE-D-22-23187R1

Dear Dr. Bennett,

We’re pleased to inform you that your manuscript has been judged scientifically suitable for publication and will be formally accepted for publication once it meets all outstanding technical requirements.

Kind regards,

Frantisek Sudzina

Academic Editor

PLOS ONE

Additional Editor Comments (optional):

Reviewers' comments:

Reviewer's Responses to Questions

**Comments to the Author**

1. If the authors have adequately addressed your comments raised in a previous round of review and you feel that this manuscript is now acceptable for publication, you may indicate that here to bypass the “Comments to the Author” section, enter your conflict of interest statement in the “Confidential to Editor” section, and submit your "Accept" recommendation.

Reviewer #1: All comments have been addressed

2. Is the manuscript technically sound, and do the data support the conclusions?

Reviewer #1: Yes

3. Has the statistical analysis been performed appropriately and rigorously? 

Reviewer #1: Yes

4. Have the authors made all data underlying the findings in their manuscript fully available?

Reviewer #1: Yes

5. Is the manuscript presented in an intelligible fashion and written in standard English?

Reviewer #1: Yes

6. Review Comments to the Author

Reviewer #1: The reviewer comments have been addressed adequately and no more changes are needed. The paper focuses on the importance of resilience in the context of COVID-19 and the authors have done a thorough presentation of their findings.

7. PLOS authors have the option to publish the peer review history of their article (what does this mean?). If published, this will include your full peer review and any attached files.

Reviewer #1: No
